# Unlocking archival maps of the Hornsund fjord area for monitoring glaciers of the Sørkapp Land peninsula, Svalbard

**Justyna Dudek[1] and Michał Pętlicki[2,3]**

[1]Institute of Geography and Spatial Organization Polish Academy of Sciences (IGSO PAS), Poland
[2]Faculty of Geography and Geology, Jagiellonian University, Cracow, Poland
[3]Department of Geography, Universidad de Concepción, Concepción, Chile

**Correspondence:** Justyna Dudek (justyna_dudek@wp.pl)

**Abstract.** Archival maps are an important source of information on the state of glaciers in polar zones and are very often basic research data for analysing changes in glacier mass, extent, and geometry. However, basing a quantitative analysis on archival maps requires they be standardised and precisely matched against modern-day cartographic materials. This can be achieved effectively using techniques and tools from the field of Geographic Information Systems (GIS). The objective of this research was to accurately register archival topographic maps of the area surrounding the Hornsund fjord (southern Spitsbergen) published by the Polish Academy of Sciences, and to evaluate their potential for use in studying changes in the geometry of glaciers in the north-western part of the Sørkapp Land peninsula in the periods: 1961–1990, 1990–2010 and 1961–2010. The area occupied by the investigated glaciers in the north-western Sørkapp Land decreased in the years 1961-2010 by 45.6 km$^2$, i.e. by slightly over 16%. The rate of glacier area change varied over time and amounted to 0.85 km$^2$/yr in the period 1961-1990 and sped up to 1.05 km$^2$/yr after 1990. This process was accompanied by glacier surface lowering by about 90-100 m for the largest land-terminating glaciers on the peninsula, and by up to more than 120 m for tidewater glaciers (above the line marking their 1984 extents).

## 1 Introduction

The Svalbard archipelago (Fig. 1a and b) is among the regions that have experienced the fastest climate warming recorded in the Arctic after the Little Ice Age (LIA) (Nordli et al., 2020). Since the beginning of the last century, the average annual temperature in this area has increased by 2.6°C per century, which is more than twice the average for other areas of the world (Nordli et al., 2014). In addition, a rapid acceleration of the pace of this process has been observed since the late 1990s (Isaksen et al., 2016). The current intensification of climate change translates into the evolution and dynamics of glacier systems, resulting in their negative mass balance and frontal recession (Nuth et al., 2010; Morris et al., 2020; Schuler et al., 2020). Changes in glacier geometry constitute a visible and easily measured parameter that, in addition to being a reliable indicator of its condition, is a proxy of changes in the natural environment (Knight, 2006).

Glaciers on Svalbard have received less attention in past research than have those in continental Europe (WGMS, 2020). This is due to their inaccessibility, harsh climatic conditions, and long polar nights, which limit the possibilities for direct measurement. Logistic and economic aspects play a crucial role in the selection of research areas, so data collection to document glacier changes (including field measurements) focuses mainly on the more accessible western coasts of the Spitsbergen island (Hagen and Liestøl, 1990). The use of traditional research methods, e.g. in situ stake mass balance measurements, is costly and time-consuming, even if the research programme is reduced to a minimum, so changes in the geometry of Svalbard glaciers are often inferred from satellite data and aerial photographs (Jacob et al., 2012; Nuth et al., 2013; Martín-Moreno et al., 2017; Girod et al., 2018;

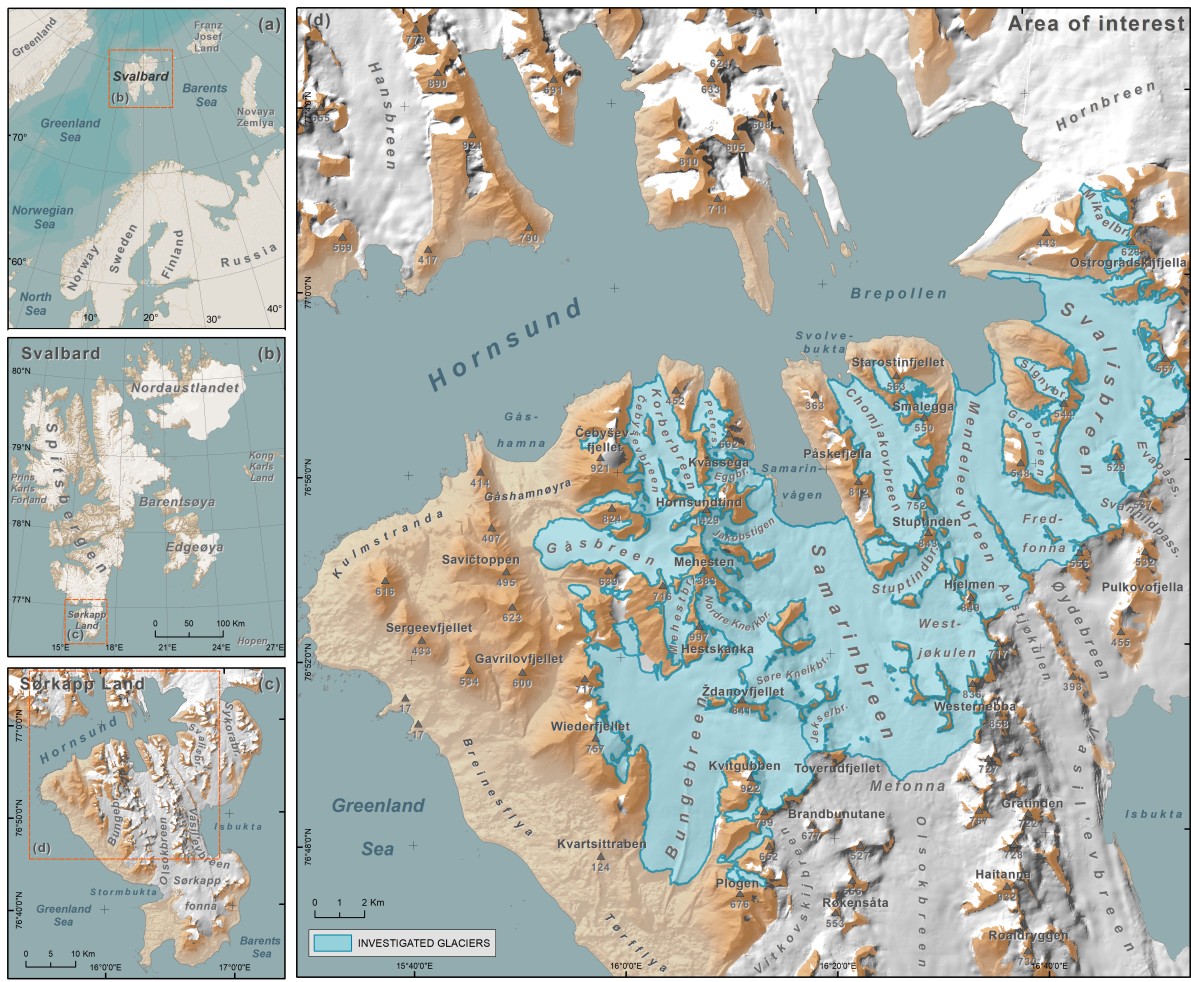

**Figure 1.** Location of the study area in the background of: (a) the Svalbard archipelago, (b) the Sørkapp Land peninsula, and (c) its north-western part. Background maps based on NPI, 2014.

Geyman et al., 2022). The use of remote sensing has a number of advantages in glacier research, the most important of which is that the data do not require a large team in the field and can be used to quickly generate precise results. These factors have certainly contributed to the use of remote sensing methods since almost the very beginning of glaciological research (Finsterwalder, 1954; Stocker-Waldhuber et al., 2019; Mannerfelt et al., 2022).

In the initial period of polar research based on remote-sensing methods, ground-based photogrammetry techniques were mainly used. In Svalbard, terrestrial photogrammetric methods were first used in 1898 as part of topographic work carried out by a Swedish expedition led by A.G. Nathorst (Nathorst, 1909). Later, these techniques were successfully used on several research expeditions organised, among others, by the prince of Monaco in 1906 and 1907 (Isachsen et al., 1912-14), and on numerous Norwegian expeditions in

1909–26 (Hoel, 1929). The aim and scientific fruit of the first photogrammetric works on Svalbard were, above all, topographic maps of poorly known areas, which were also valuable material for the study of glacier extents. Polish achievements in this field include a series of photogrammetric images and triangulation measurements made in 1934 as part of the first Polish research expedition to the as-yet-unexplored Torell Land (southern Spitsbergen), which yielded the first map of this area at a scale of 1:50,000 (Zagrajski and Zawadzki, 1936).

The construction of the Polish Polar Station on Isbjørnhamna Bay in 1957 allowed scientific teams to operate in southern Spitsbergen. In the first years of the station, a Polish research team led by C. Lipert conducted terrestrial photogrammetric measurements, resulting in the production of detailed maps of glaciers in the vicinity of the Hornsund fjord (Kosiba, 1960; Lipert, 1962). Additionally, topographic

sketches of the Antoniabreen and Penckbreen glaciers were made during an expedition to the vicinity of the Van Keulen fjord in the same period (Marcinkiewicz, 1961). Changes in the extent of glaciers around the station were also docu-
5 mented in the early 1970s, when summer expeditions of the University of Wrocław were held there (Żyszkowski, 1982), and, after activity resumed in 1978, on numerous expeditions carried out mainly by the University of Silesia and the Insti-tute of Geophysics of the Polish Academy of Sciences (*Insty-*
10 *tut Geofizyki, Polska Akademia Nauk*, hereinafter referred to as IGF PAN) (Kolondra, 2000). Terrestrial photogrammetric methods are still used today in glaciological studies of this area, and the longest series of measurements have covered the Werenskioldbreen, Torellbreen, and Hansbreen glaciers
(Kolondra, 2002).

Compared to other areas of Svalbard, the photogrammetric research on the Sørkapp Land peninsula and the number of related cartographic works published are very modest. The area most often chosen for cartographic studies has been the
20 north of the peninsula (which is relatively accessible from the Hornsund fjord), including mainly the Gåsbreen area (De Geer, 1923; Pillewizer, 1939; Jania, 1979, 1982; Kolondra, 1979, 1980; Schöner and Schöner, 1996, 1997; Ziaja et al., 2016) and the glaciers that flow into the fjord (Heintz, 1953;
Błaszczyk et al., 2013).

Terrestrial photogrammetric methods provide precise and repeatable results. However, they can only be used for small-scale studies, usually covering one glacier or its foreland (Kolondra, 2005). For glaciological studies with extensive
spatial coverage, data obtained from the aerial ceiling are much more competitive providing information from large and hard-to-reach areas, which is of great importance in po-lar conditions. Therefore, aerial photogrammetry progressed along with ground measurement techniques on Svalbard.
Professional photogrammetric overflights by the Norwegian Polar Institute (NPI) covered the entire or almost all of the Sørkapp Land peninsula (Table 1). The first, in 1936, re-sulted in a series of oblique photos that were used to create a 1:100,000 topographic map that covered the entire Svalbard
archipelago (Luncke, 1936; NPI, 1986; Geyman et al., 2022). Another map by the Norwegian Polar Institute was published only in 2007, and was based on 1:50,000 vertical photos from 1990, this time as colour prints (NPI, 2007). For the study of glacier evolution and glacial landforms on the Sørkapp Land
peninsula, the series of photos taken in 1961 is of great im-portance because, for the first time in the history of this area, it uniformly covered all its glaciers along with their marginal zones. Another data set of the same spatial extent has not been created until 2010 (Fig. 2).
Norwegian photos from two photogrammetric campaigns in 1960 and 1961 have served as the source material for many cartographic and glaciological works (e.g. Klysz and Lind-ner, 1982; Ostaficzuk et al., 1982; Jania, 1987, 1988a, b; Schöner and Schöner, 1996; James et al., 2012; Błaszczyk et
al., 2013; Małecki, 2013). Of the available cartographic stud-

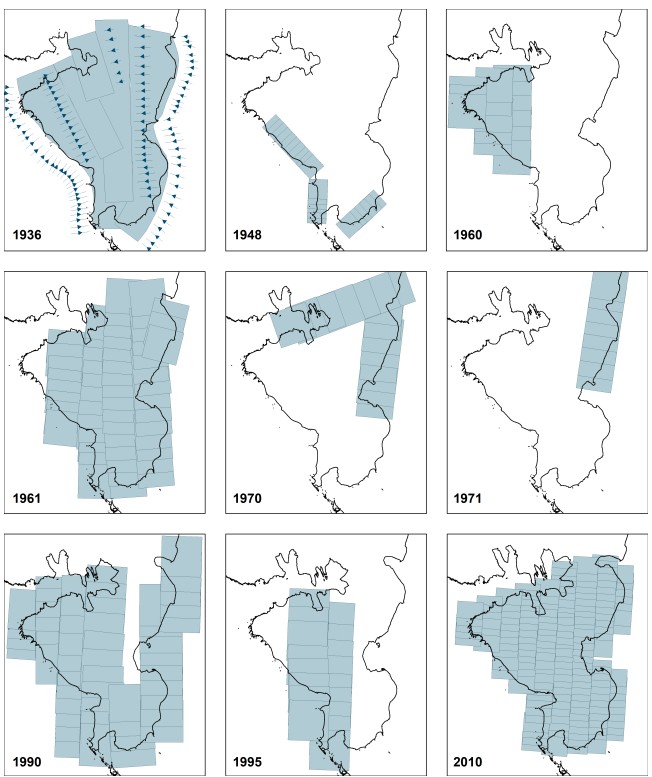

**Figure 2.** Norwegian Polar Institute photogrammetric campaigns carried out over the Sørkapp Land peninsula.

ies valid for 1960/61, the 1:25,000 topographic map of the Hornsund fjord area has the greatest spatial coverage (1600 km$^2$). The series of sheets published in 1987 was, in part, the result of the IGF PAN programme of expeditions to Spits-bergen in the years 1979-1984 with the support of officers of 60 the Polish military cartographic institute (*Wojskowe Zakłady Kartograficzne*) in conducting desk research and field work. Field survey reference photogrammetric measurements were made during the 6th expedition of the Polish Academy of Sciences in 1984. The present study aims to assess its accu- 65 racy and its potential for use in research on changes in glacier geometry on the north-western Sørkapp Land peninsula.

The use of archival cartographic data is one of the key ways to quantify mainly climate change-related changes in the cryosphere (Nuth et al., 2007, 2013; Surazakov et al., 70 2006; Weber et al., 2020; Holmlund, 2021). On a global scale, such data on the topography of glaciers from the 1960s are relatively scarce – they are mainly based on a few pho-togrammetric surveys and the resultant topographic maps (Tielidze, 2016; Andreassen et al., 2020) and a reanalysis 75 of declassified spy satellite images (Bhambri et al., 2011; Bhattacharya et al., 2021; Maurer et al., 2015, 2019). In the Spitsbergen region, 1930s surveys are a key reference point for observed changes in area and volume (Nuth et al., 2007). Structure-from-motion (SfM) photogrammetry allows better 80 and more precise use of these photos and the creation of more

**Table 1.** Norwegian Polar Institute photogrammetric campaigns carried out over the Sørkapp Land peninsula.

| Year | Area of Sørkapp Land covered | References |
|---|---|---|
| 1936 | Entire peninsula | Błaszczyk et al. (2013), Dowdeswell et al. (1995), Geyman et al. (2022), Hagen et al. (1993), Heintz (1953), Jania (1988a, b), Jiskoot et al. (2000), König et al. (2014), Lefauconnier and Hagen (1991), Luncke (1936), Martín-Moreno et al. (2017), Noormets et al. (2020), NPI (1948, 1986, 2014), Nuth et al. (2007, 2013), Pälli et al. (2003), Sharov (2006) Sharov and Osokin (2006), Sund et al. (2009), Szafraniec (2018, 2020), Ziaja (2001, 2004); Ziaja et al. (2007, 2009); Ziaja and Ostafin (2015), |
| 1948 | Western and southern coasts | – |
| 1960 | Western part of the peninsula | Błaszczyk et al. (2013), Hagen et al. (1993), Jania (1987, 1988a, b), Schöner and Schöner (1996, 1997), Ziaja (2004), |
| 1961 | Almost entire peninsula except the NW | Barna and Warchoł (1987), Grabiec et al. (2017), Jania (1987, 1988a, b), Jania and Szczypek (1987), Klysz and Lindner (1982), Lefauconnier and Hagen (1991), Noormets et al. (2020), Ostaficzuk et al. (1982),Schuler et al. (2020), van Pelt et al. (2019, 2021), Ziaja (2004); Ziaja et al. (2007, 2016), |
| 1970 | Isthmus and eastern coast | Dowdeswell et al. (1995), Lefauconnier and Hagen (1991), Noormets et al. (2020), Nuth et al. (2013), NPI (2014), Schuler et al. (2020), van Pelt et al. (2019, 2021), Ziaja (2004) |
| 1971 | Eastern coast | Dowdeswell et al. (1995), Lefauconnier and Hagen (1991), Ziaja (2004); Ziaja et al. (2007, 2009) |
| 1990 | Almost entire peninsula except the NE | Błaszczyk et al. (2013), Fürst et al. (2018), Jiskoot et al. (2000), König et al. (2014), Noël et al. (2020), NPI (2014), Nuth et al. (2007, 2010, 2013), Schöner and Schöner (1996, 1997), Schuler et al. (2020), Sund et al. (2009), Szafraniec (2020), van Pelt et al. (2019, 2021), Ziaja (2004) Ziaja et al. (2007, 2016); Ziaja and Dudek (2011); Ziaja and Ostafin (2015), |
| 1995 | Central part of the peninsula | NPI (2014) |
| 2010 | Entire peninsula | Farnsworth et al. (2016), Fürst et al. (2018), NPI (2014), Ziaja et al. (2016), |

accurate elevation models (Mertes et al., 2017; Midgley and Tonkin, 2017; Girod et al., 2018; Holmlund, 2021; Geyman et al., 2022). Unfortunately, high-quality scans of aerial imagery from 1961 that could be used to generate new DEMs using stereophotogrammetry or structure-from-motion methods were not available for this study. An attempt was made to process the possessed low-resolution scans with Agisoft Photoscan; however, the results were not satisfactory. The primary focus of the study was on the use and recovery of historical maps.

## 2 Study Area

The Svalbard archipelago is surrounded by the Greenland Sea to the west, the Barents Sea to the east, and the Arctic Ocean to the north. The temperature of the bordering water masses influences its climate, which is milder than that of other areas at similar latitudes and, at the same time, more sensitive to changes related to the passage of atmospheric fronts (Hagen et al., 1993; Eckerstorfer and Christiansen, 2011). The East Spitsbergen Current transports cold, Artic Waters along the eastern shores of the archipelago, whereas the West Spitsbergen Current, a branch of the Gulf Stream, brings warm Atlantic waters to the western shores. The resulting strong climate gradients cause a pronounced latitudi-

nal and longitudinal variability in the ice cover of Svalbard, with the central part of Spitsbergen being largely ice-free due to low precipitation and the eastern shores being more glaciated than the western. Many of the Svalbard glaciers have surged in the recent past (Farnsworth et al., 2016) or are currently undergoing an active surge phase (Sund et al., 2009), with a substantial short-term increase in ice flow velocity. The Sørkapp Land is the southernmost peninsula of Spitsbergen, the largest island of the Svalbard archipelago (Fig. 1a and b). It is separated from the rest of Spitsbergen by the Hornsund Fjord and a narrow glaciated isthmus of Hornbreen-Hambergbreen (Ziaja and Ostafin, 2015). There is ongoing speculation whether Sørkapp Land will form a separate island when the ice in the isthmus is gone (Pälli et al., 2003; Grabiec et al., 2017). Compared to the rest of Svalbard, Hornsund, and Sørkapp Land have a mild and humid climate (Isaksen et al., 2016). Meteorological measurements were not carried out until the 1970s in the southern Spitsbergen. According to Førland et al. (2011) the mean annual temperature at the Svalbard airport in the period of 1966-1988 increased by 0.52°C/decade, and in the following period (1988-2011) this trend continued with an increase of the mean annual temperature to 1.25°C/decade. In recent decades, a similar trend was observed in southern Spitsbergen, where a pronounced increase in winter air temperature

and summer precipitation sums was observed at the Polish Polar Station Hornsund where in the period 1971-2000, the mean annual temperature was -4.7°C which in the following years (2001-2015) increased by 1.9°C (Førland et al., 2011; Osuch and Wawrzyniak, 2017).

The north-western Sørkapp Land region extends between the open Greenland Sea and Hornsund Fjord (Fig. 1c and d). It hosts 20 land-terminating glaciers, 8 tidewater glaciers, several rock glaciers, and numerous glacierets and perennial snow patches. The largest land-terminating glaciers in the analysed area are Gåsbreen, which is surrounded by the highest mountain massifs of southern Spitsbergen and fed by the Bastionbreen and Garwoodbren tributary glaciers, both of which rest on the slopes of the Hornsundtind massif; and Bungebreen, which extends meridionally between the high Hestskanka massif to the north and the Tørfflya coastal lowland to the south. The two largest glacial systems in western Sørkapp Land are surrounded by smaller valley and cirque glaciers. East of Samarinvågen Bay, there are also several smaller land-terminating glaciers that constitute former tributaries of larger glaciers flowing into the Hornsund fjord.

Of the 16 glaciers flowing into the Hornsund Fjord, 8 are located in the study area. These glaciers have a northern exposure and their snouts move northwards. Some of them originate on glaciated mountain passes in the interior of the peninsula where they share their accumulation zones with other glaciers. Samarinbreen, which flows into Samarinvågen, one of the bays in the Hornsund Fjord, is the largest tidewater glacier in the region. It flows from the Mefonna Ice Plateau, where it connects to the Olsokbreen glacial system that flows into the Greenland Sea. To the east, Samarinbreen is adjacent to the glaciers feeding the Vasil'evbreen glacier basin, and to the west to the wide accumulation zone of the Bungebreen glacier. Samarinbreen is a compound glacier basin fed by numerous tributary glaciers: Westjøkulen and Stuptindbren, flowing westward from the slopes of Westernebba, Hjelmen, and Stuptinden; as well as flowing eastward tributaries: Jekselbreen, Søre Kneikbreen, Nordre Kneikbreen, Jakobstigen situated in depressions between the Toverudfjellet, and Horsundtind massifs.

Further east of Samarinbreen are the Mendeleevbreen and Svalisbreen that flow into Brepollen Bay. Both glaciers have a broad connection to the adjacent basin of Vasil'evbreen. Mendeleevbreen flows from the Austjøkulen ice plateau, and it receives additional supply from the Fredfonna ice plateau and Grobreen. The neighbouring Svalisbren occupies a depression with an atypical sinusoidal course. The main accumulation zone of the glacier is located on the Svanhildpasset.

In the depression between the Påskefjella and Smalegga massifs, there is the fourth largest glacier in the region - Chomjakovbreen, which fills a separate valley and does not connect with other glaciers. It calves with a not very wide cliff to the small Svovelbukta bay. It is fed by small but very numerous tributary glaciers located in cirques on the slopes

of the surrounding mountain massifs. The longest tributary that feeds it is Dmitrievbreen.

In addition to the vast compound glacier basins, a number of smaller valley glaciers also flow into the Hornsund Fjord. These include Körberbreen with Čebyševbreen tributary, as well as Petersbreen, Kvasseggbreen, and Eggbreen glaciers, which are to the west of Samarinbreen, where they fill deep valleys. They are distinguished from other outlet glaciers by their significant vertical range and associated steeper surface, which is due to the fact that their basin boundaries run along the highest mountain ranges of southern Spitsbergen: Čebyševfjellet (914 m a.s.l.), Wesletinden (928 m a.s.l.), Hornsundtind (1,429 m a.s.l.) and Kvasegga (1,004 m a.s.l.) (Jania, 1987). Körberbreen and Petersbreen lie in separate longitudinal mountain valleys whose depth and direction are determined by the geological structure of the substrate, which relates to the course of faults. The two small glaciers Kvasseggbreen and Eggbreen, which run adjacent to them to the east, run latitudinally and flow into Samarinvågen Bay. They formerly served as the tributary glaciers to Samarinbreen, but as its snout has receded, they split from it and today constitute separate calving glaciers.

## 3  Source Material

The basis for the spatial analysis consisted primarily of topographic maps and DEMs, supplemented with aerial photos and satellite images. A specification of these datasets is provided below.

### 3.1  Maps

### 3.1.1  IGF PAN topographic map series

The IGF PAN topographic map series, made in a Universal Transverse Mercator (UTM) projection (northern hemisphere, zone 33) based on a European Datum 1950 (ED50) ellipsoid, consisted of ten sheets. This study assessed six of those sheets that represented glaciers with adjacent marginal zones and one sheet representing ice-free area in the territory of Sørkapp Land peninsula (No. 3 - Hornbreen, No. 5 – Hornsund, No. 6 - Brepollen, No.7 - Pällfyoden, No. 8 – Gåsbreen, no. 9 - Samarinbreen, and No. 10 – Bungbreen; Fig. 3).

The topographic map sheets presented the relief; permanent and periodic watercourses; water bodies; wetlands; glaciers; triangulation and topographic points; vegetation types (tundra); marine coasts (skerries); and names of geographical features (Fig. 3). The relief is presented using contour lines with contour intervals of 5 m for the relatively flat coastal plains and 10 m for steeper areas. Areas too steep to be mapped using contour lines in the assumed scale were presented as rock cliff symbols. The extents of land-terminating glaciers are marked as a change in contour line colour from orange (land) to blue (glacier), but lines were not

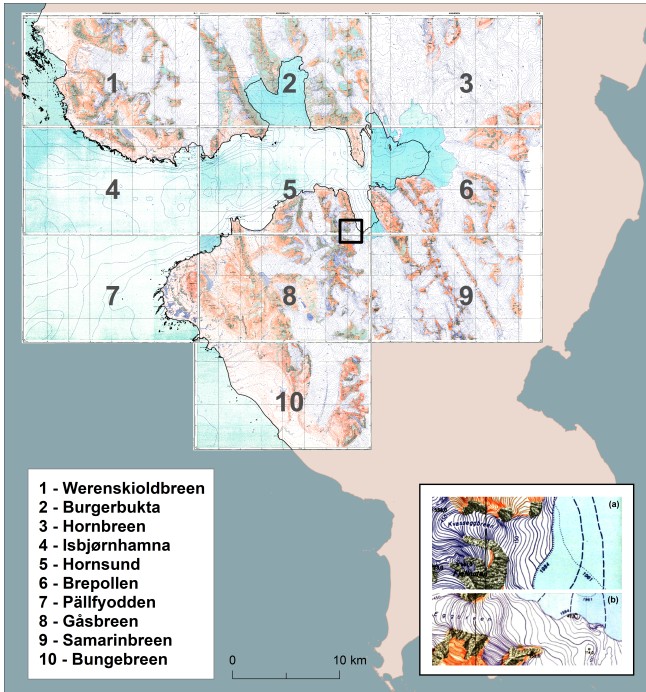

**Figure 3.** IGF PAN topographic map series published in 1987 (Barna and Warchoł, 1987). Example visualisations of the extents of tidewater glaciers on IGF PAN maps published in 1987: (a) Kvasseggbreen (Sheet 5 – Hornsund); (b) Eggbreen (Sheet 8 – Gåsbreen).

drawn to mark the maximum extent of glacial snouts. Two extents are marked for tidewater glaciers (the Petersbreen, Kvasseggbreen, Eggbreen, Smarinbreen, Chomjakovbreen, Mendeleevbreen, and Svalisbreen). The first – a dotted line
on the surface of the Hornsund Fjord – showed the position of their termini in 1961. The second extent, represented as ice cliffs, was the 1984 update of their boundaries (Fig. 3a and b).

When individual sheets were converted to digital form, it
was important to take into account the particularity of the maps that resulted from the means by which they were created. Initial photogrammetric sketches of individual sheets were made using the 1961 Norwegian aerial photos from before the expeditions to Spitsbergen of the early 1980s (in-
cluding the expedition to Sørkapp Land in the summer of 1984 – verbal communication: W. Ziaja). The cartographic material constituted a base that, according to the information provided in the map description, was 'partially checked and supplemented in the field'.
Information on the degree to which documentation of the extent and elevation of Sørkapp Land glaciers was updated in the field was key in assessing the potential of this series for use in analysing changes in glacier thicknesses in the periods 1961-1990-2010. In the context of this analysis, the most
important question was whether the contour lines marking

the elevation of the glaciers represent the year 1961 (which would result from the use of aerial photographs from that period) or 1984 (which would result from the contour lines having been updated using field measurements made more than two decades after the photogrammetric overflight). The com-
30 parison of the 1987 series of maps with other cartographic studies of the area presenting the state of glaciers in the early 1960s led to the unequivocal conclusion that the contours contained therein represent the year 1961 (and thus were not corrected based on field research), while their updating (by
35 'in-field supplementation') to reflect the 1984 state of affairs related only to glacier extents, as reflected in the change in contour colours.

One such study was a report from an expedition by Austrian scientists Monika and Wolfgang Schöner, who in 1991
made ground-based photogrammetric measurements on the forefield of the Gåsbreen glacier. These studies were based on photos from NPI's photogrammetric overflight over the west of Sørkapp Land in the summer of 1960 (Table 1) and resulted in a publication that included a map showing the
hypsometric variation of the Gåsbreen and a hillshade that was valid for 1960 (Schöner and Schöner, 1996). A helpful publication for comparisons of the elevations of Körberbreen and Petersbreen was an article by Jania (1987) that included hypsometric profiles of both of these glaciers valid for 1960.
Another important cartographic study was a 1:10,000 map of the forefield and lower part of the Bungebreen glacier snout by Warsaw geologists based on aerial photographs from 1961 (Ostaficzuk et al., 1982; Dzierżek et al., 1991).

### 3.1.2 NPI map
The topographic map of Sørkapp Land at the scale of 1:100,000 based on infrared aerial images from 1990 was developed by NPI and released in analogue form in 2007. The map presented the general image of the area: relief, permanent watercourses, lakes, glaciers, and elevation points.
Since the coverage of the peninsula by image data in the year 1990 was incomplete in this first dataset (Fig. 2), the gap in the north-eastern part of the peninsula was filled by the data from 1961 (Fig. 4). In 2014, NPI launched a geoportal (data.npolar.no) enabling spatial data viewing, down-
loading, and processing. The first online map of the Sørkapp Land peninsula (C13) was available in shapefile format. In the study, the vector layers that present glaciers and elevation points were used.

### 3.2 Imagery
The image data, consisting mainly of aerial photos captured during the photogrammetric overflights commissioned by the NPI, were used to delineate the extents of the glaciers, as well as visual inspection of the reference DEMs (Fig. 5). The data for 1961 included scans of fourteen vertical aerial
photos from the historical photogrammetric campaign over

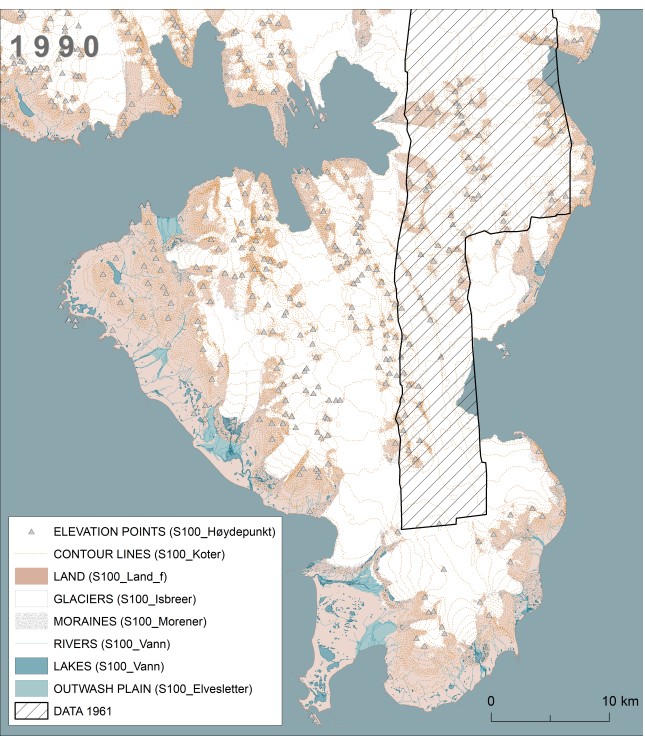

**Figure 4.** Topographic map of the southern Spitsbergen published in 2007 and released online by NPI (2014).

the Sørkapp Land area on August 24 and 25, 1961 (Fig. 5a). Black-and-white images at a scale of 1:50,000, subjected to photointerpretation, were captured from a ceiling of about 8,000 m using a Wild RC camera with a focal length of
153.45 mm (Jania, 1987).

A basis for the delineation of glaciers' boundaries in 1990 by the NPI consisted of infrared aerial photos at a scale of 1: 50000 registered by the RC-10 camera with a focal length of 152 mm. Two stripes for the north-eastern part of the penin-
10 sula were missing in the set (Fig. 1); thus, in this study the outlines of four glaciers (Svalisbreen, Sigygbreen, Grobreen, and Mikaelbreen) were delineated based on the image acquired on August 20, 1990, by the Landsat 5 Thematic Mapper (TM) sensor (Fig. 5b).

The last photogrammetric campaign carried out by the NPI in 2010, covered the entire peninsula (Fig. 1). Photos with a resolution of 0.4 m were captured on August 17 from a ceiling of about 7350 m a.s.l. by the multispectral digital camera UltraCam Xp with a focal length of 100.5 mm. For glaciers'
delineation in 2010 in this study, we used an orthoimage released in 2020 by the NPI (Fig. 5c).

### 3.3  DEMs

A baseline elevation dataset used for the analysis of map accuracy and changes in glacier thickness consisted of Digi-
25 tal Elevation Models (DEM) elaborated by the NPI (2014).

This study employed a 5-m-resolution DEM generated from digital images captured in 2010 and two DEMs with 20-m-resolution based on archival aerial photographs acquired by the NPI in 1990 and 1961 (Fig. 6). DEMs were generated using photogrammetric methods based on stereopairs cor-  30 relation. The vertical accuracy of the DEMs given by the NPI was 2-5 metres in non-glacial areas and slightly lower for glacier surfaces (NPI, 2014). The model representing the year 2010, defined by the author as the most accurate, was chosen as a reference dataset throughout the study, and the  35 older models were resampled at a 5-m-resolution, which allowed further compilation with other spatial data.

## 4  Methods

### 4.1  Source data processing and evaluation of output data accuracy  40

#### 4.1.1  Data for 1990

Evaluation of data accuracy produced by NPI for the year 1990 was done by checking the extent to which it fitted existing reliable altitude data for areas not subject to large natural changes over time (in practice, this was the majority of non-  45 glaciated areas). The most reliable source of data for comparisons was the 2010 DEM generated by NPI using photogrammetric methods based on aerial photos and field-measured control points (NPI, 2014). The model from 1990 was validated for horizontal shift against the reference dataset using a  50 procedure developed by Nuth and Kääb (2011) that proposed an analytical solution of a 3-dimensional shift vector between two DEMs, by relating the elevation differences to the elevation derivatives of slope and aspect. The correction process was done iteratively until the magnitude of the shift vec-  55 tor approached zero. This method was applied for a terrain considered stable, that did not experience changes in geometry over time, and when the values of the shift vector were solved, a correction was applied to the whole DEM. Before the correction of the DEM from 2010 versus the DEM from  60 1990 the magnitude of the shift was 2.14 m in the eastern direction. The mean elevation bias between compared data (z shift component) before the correction was 2.13 m (Fig. 7).

#### 4.1.2  Data for 1961

The maps on which the glacier elevation analysis was based  65 in 1961 in the northwest of Sørkapp Land were processed in several steps using ESRI ArcGIS and Matlab software (Fig. 8). The analogue maps were first scanned and converted to *TIFF* format, then defined in the UTM projection, based on a European Datum 1950 (ED50) ellipsoid, in which  70 the background maps had been developed. Subsequently, the coordinate system was converted and the UTM projection was adopted into the ETRS 89 reference system. This allowed cartographic compilation and integration with other data used for spatial analyses.  75

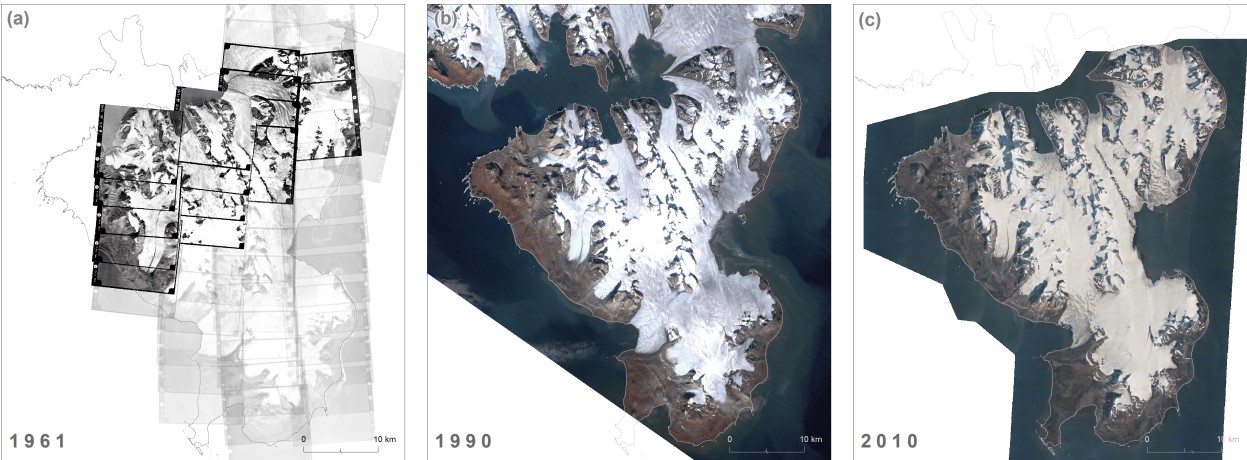

**Figure 5.** Image data covering Sorkapp Land used in this study: (a) aerial photographs from 1961 taken on August 24 and 25, 1961; (b) Landsat TM5 image captured on August 20, 1990; (c) orthophoto created by the NPI (2014) from digital photos captured on August 26, 2010.

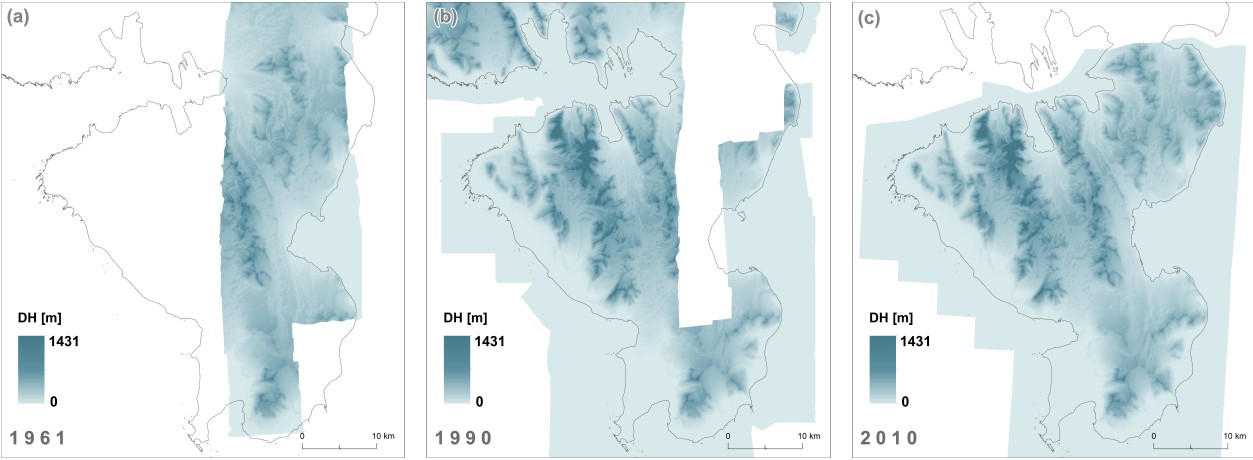

**Figure 6.** DEMs covering Sørkapp Land generated from aerial photographs by the NPI (2014)

Georeferenced maps were used to generate contour lines and vector layers showing the hydrographic network (rivers and lakes), as well as peaks and other elevation points. For this purpose, the raster cleanup and automatic vectorisation functions available in the Arc Scan extension were used. This tool proved very useful for converting raster maps to vector format, as it allowed for significantly faster digitisation of contour lines (for western Sørkapp Land, with its very diversified relief, they were very densely packed – every 5 or 10 m – in the altitude range from 0 up to 1,430 m a.s.l.). In the next step, control over the quality of the final result was carried out, and the generated features were manually edited and merged (Fig. 9). Georeferenced GIS layers – contours, peaks, and topographic points – were supplemented with information about the elevation in the attribute table. Subsequently, contours were converted to elevation points with a

5 m spatial distance along the lines and merged with layers of peaks and topographic points.

The final step was to verify the relative accuracy of the elevation data obtained. The model representing the year 2010 was chosen as a reference dataset for comparisons and data validation. The easiest way to verify the differences between the two datasets was to subtract one from the other and then interpolate the difference map as proposed by McNabb et al. (2019). A layer of the elevation difference between the datasets was saved as a shapefile that was subsequently used to generate a Triangulated Irregular Network(TIN). In the next step, this model was transformed into a regular GRID at a spatial resolution of 5 m. The DEM of Difference (DoD) obtained is shown in Figure 10a.

The first result of the comparison was not satisfactory. In the western part of the DoD, there were large negative values on slopes with eastern exposure, along with large posi-

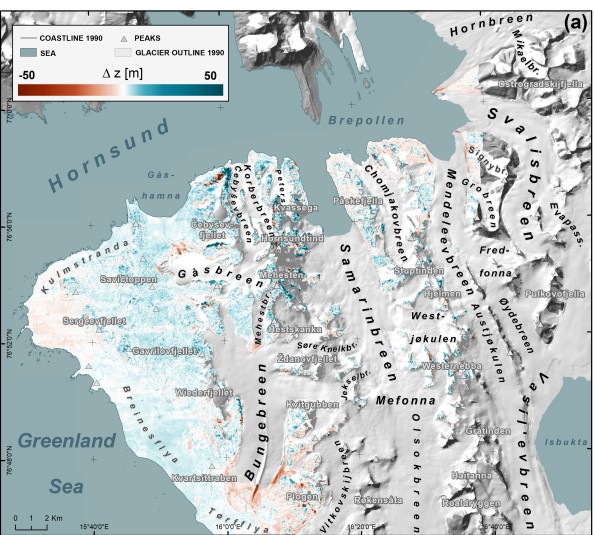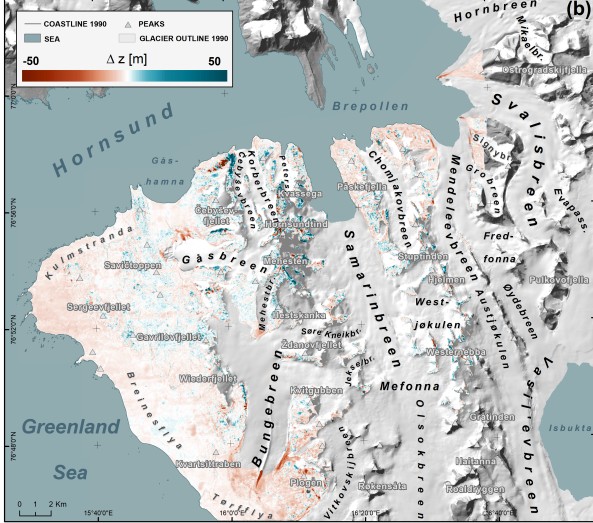

**Figure 7.** Elevation differences between the 1990 DEM and the 2010 DEM generated by NPI (2014) (a) before applying correction (b) after correcting for shift vector as proposed by Nuth and Kääb (2011).

**Table 2.** Shift vector between data from 1961 and 2010 calculated using a procedure developed by Nuth and Kääb (2011).

| Map Sheet | Land (km$^2$) | Mask (km$^2$) | Mask (%) | Shift (m) | | |
|---|---|---|---|---|---|---|
| | | | | x | y | z |
| 3 | 18.5 | 3.0 | 16.2 | -7.59 | 10.60 | -2.42 |
| 5 | 29.6 | 9.8 | 33.1 | -32.40 | -5.68 | 0.57 |
| 6 | 119.9 | 24.2 | 20.2 | -28.13 | 27.66 | -2.75 |
| 7 | 15.8 | 11.2 | 71.0 | 3.02 | 8.19 | -0.96 |
| 8 | 155.4 | 65.3 | 42.1 | -46.98 | 19.77 | -1.44 |
| 9 | 143.4 | 14.2 | 9.9 | -24.76 | 23.24 | 3.38 |
| 10 | 113.7 | 63.6 | 55.9 | -31.61 | 32.72 | -2.85 |

tive values on western slopes, indicating that the two datasets were horizontally offset in relation to each other (Nuth and Kääb, 2011).

In order to define horizontal shift between two datasets a procedure developed by Nuth and Kääb (2011) was applied to each sheet separately. The results, as presented in Table 2 and Figure 10b, indicated that the shift vector wasn't uniform for all maps in the series. Moreover, although the algorithm permitted improvement of the position of data for 1961 and diminish the horizontal errors, on the peripheries of each map sheet the shift between both datasets was still indicated by elevation differences on the slopes (Fig. 10b).

In view of this result, it was attempted to estimate the position errors of the IGF PAN vector layers and then correct them. In the first stage, the locations of the elevation points were assessed. The analysis was performed for each of the sheets separately. The maps shared 189 points representing the same places, of which 27 were triangulation points. The remaining elevation points were mainly peaks, but some indicated geographical features in the field. For the purposes of this study, both types of points were assigned to a common category of objects called "topographic points" (Fig. 9). Furthermore, several topographic points (including points showing the position of building Camp Erna and Konstantinovka) were added in order to match both datasets. Differences in their position in relation to each other are presented in the Supplement Table.

From the IGF PAN map series two sheets (No. 8 - Gåsbreen and No. 9 - Samarinbreen) covered the interior of the study area, the adjacent further two (No. 6 - Brepollen, 10 - Bungebreen) showed less land, and the remaining two (No. 3 - Hornbreen, No. 5 - Hornsund) were on the peripheries of the study area. Below is the description of each used map sheet in ascending order.

The map sheet No. 3 - Hornbreen, covering the northernmost part of the study area adjacent to the glaciated isthmus, overlapped with the data for 2010 only to a small extent, Figure 9 shows the southern part of the map that was useful in this study and the 3 points on which the sheet registration was based.

Similarly for the next IGF PAN map sheet (No. 5 – Hornsund), showing the north-western peripheries of the study area, only the southern part was used. The map sheet contained 18 elevation points which constituted the basis for registration. Figure 9 shows their position and distribution.

The map sheet No. 6 - Brepollen showed areas further east with Brepollen bay and the fronts of three large tidewater glaciers. On this sheet 34 points were used to match the NPI dataset. Due to the difficult accessibility of this area, only one of them constituted a triangulation point (Fig. 9).

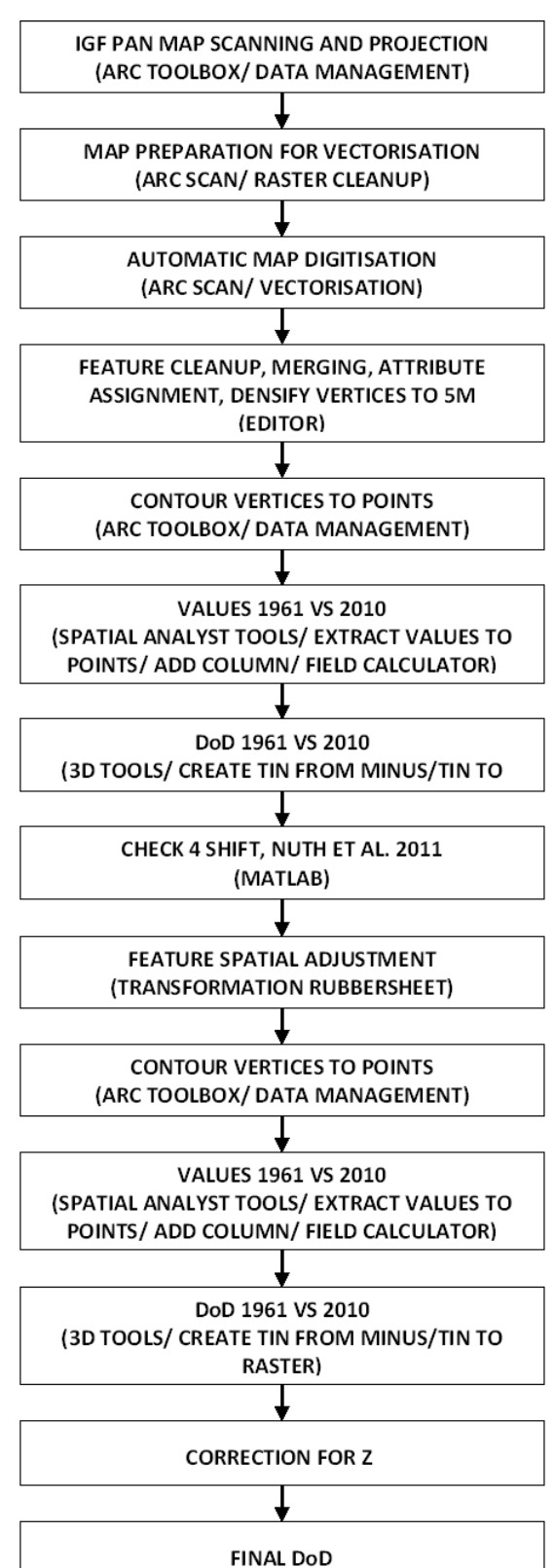

**Figure 8.** Work flow for processing IGF PAN map sheets.

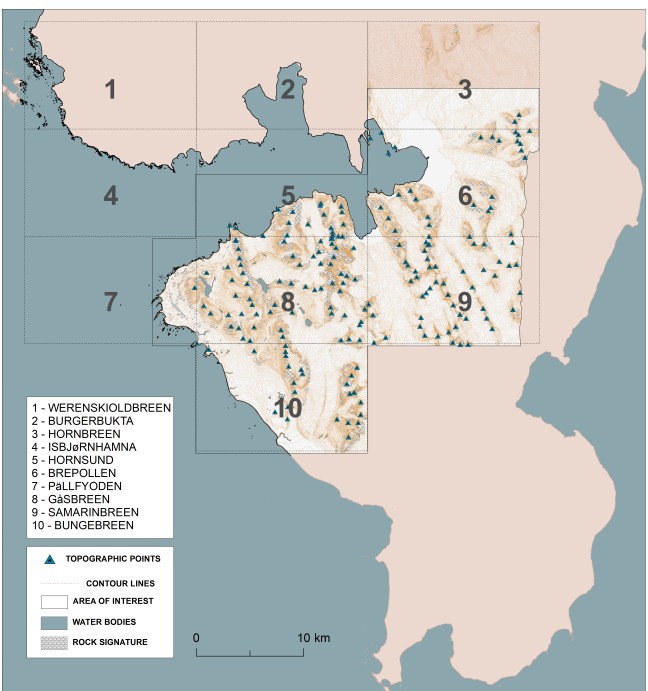

**Figure 9.** Contours and topographic points of map sheets used in the study.

The No. 8 - Gåsbreen map sheet, which covers the largest area of the peninsula, contained 85 elevation points, which were used for map registration (Fig. 9). Comparing the location of individual points, it can be concluded that the 1961 Gåsbreen map sheet was shifted south-eastward relative to the 2010 NPI data. The peaks of the mountain massifs around Hornsundtind were reproduced the most accurately. Moving westward from Hornsundtind, the distance between the topographic points on both maps increased, which led to the assumption that this was not a simple shift between maps, but rather that the problem is a distortion resulting from, among other things, coordinates on the mapping grid being marked incorrectly. The possibility of this problem was already indicated in the description of the IGF PAN map sheets, which explained that the UTM geographic coordinates obtained from NPI that it used differ from the geographic coordinates obtained from astronomical measurements (Barna and Warchoł, 1987).

The next map sheet - No. 9 - Samarinbreen showed the adjacent areas to the east – a large tidewater glacier Samarinbreen and the accumulation zones of Mendeleyevbreen and Svalisbreen. The glaciers were separated by very steep mountain ranges with numerous peaks. The map sheet included 47 elevation points distributed relatively evenly, which could be used for registration. All the points on the map sheet were in the category of topographic points (Fig. 9).

The last corrected sheet of the 1961 map – No. 10 - Bungebreen – showed much less land, and hence fewer elevation

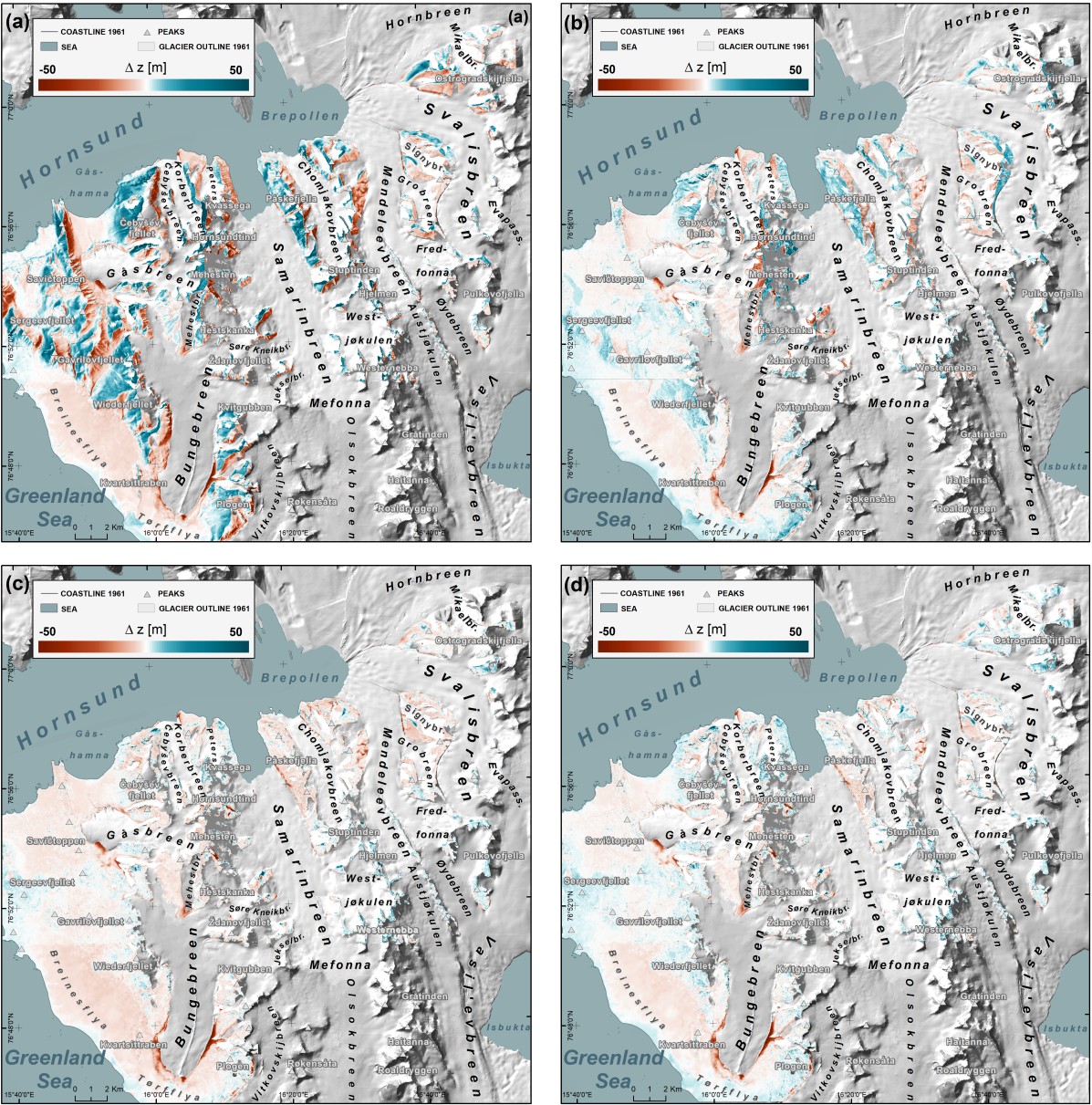

**Figure 10.** Elevation differences between the 1961 vector layers generated from IGF PAN maps issued in 1987 and the 2010 DEM generated by NPI (2014), (a) map rectification based on the nodes of the cartographic grid; (b) DEM corrected using method by Nuth and Kääb (2011); (c) data rectification based on elevation points (d) final DoD

points (Fig. 9). A preliminary assessment of map quality determined a shift in the topographic points layer relative to the contour lines, which most likely occurred while the map was being prepared for printing. To solve this problem, before registering the sheet under development, the two digitised layers were matched against each other so that the elevation points fell within the contours delineating the summits (Fig. 9).

In addition to the small number of elevation points and their shifting relative to contour lines, the planned map registration was further hampered by the uneven distribution of elevation points within the sheet. Most of the points were located on the peaks of mountain massifs in the northern and eastern parts of the map, while the points in the coastal zone in the west were missing. The corresponding portion of the map issued by the NPI for 1990 contained one topographic point at the base of Cape Rafenodden at an altitude

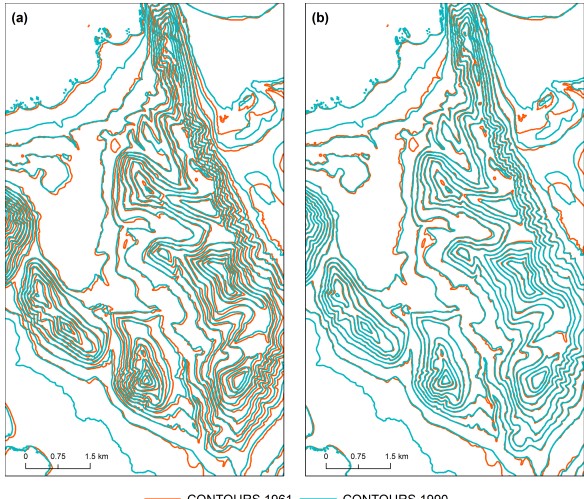

**Figure 11.** Course of contour lines in the western part of sheet 8 – Gåsbreen: georeference based on: (a) grid nodes; and (b) elevation points.

of 17 m a.s.l. To match the 1961 and 1990 data, one point was added to the Bungebreen sheet, within a small elevation delimited by a contour at 17.5 m a.s.l. In the next step, the vector layer of topographic points for 1961 was made denser by adding a few points at the peaks of four massifs. These were points within contours delineating the summits of Arkfjellet, Plogen, Wiederfjellet, and Stupprygen. The Supplement Table shows the coordinates of all points on which the registration of sheet 10 – Bungebreen was based.

## 4.2 Fitting data from 1961 and 2010

To align the 1961 vector layers with the 2010 data, they were registered (*Spatial Adjustment* function/*Rubbersheet* conversion), this time based on triangulation and topographic points (Supplement Table). The vector data thus processed were then used to generate a DoD with a resolution of 5 m, similarly as in the first step of data processing (Fig. 8).

A preliminary visual analysis of the DoD obtained (Fig. 10c) led us to conclude that a significant improvement had been achieved in terms of the spatial fit of the models. This was also indicated by a visual assessment of the comparative courses and positions of the 1961 and 1990 contours (Fig. 11-13). Considering the limited possibility of accurately determining the elevation points on which the data registration for 1961 was based, the results of comparing both vector layers and both elevation models were considered satisfactory.

## 4.3 Final elevation model for 1961

After correcting all vector data that were based on the IGF PAN map sheets and the DoD processed from them, the accuracy of the end product and its adjustment to the remain-

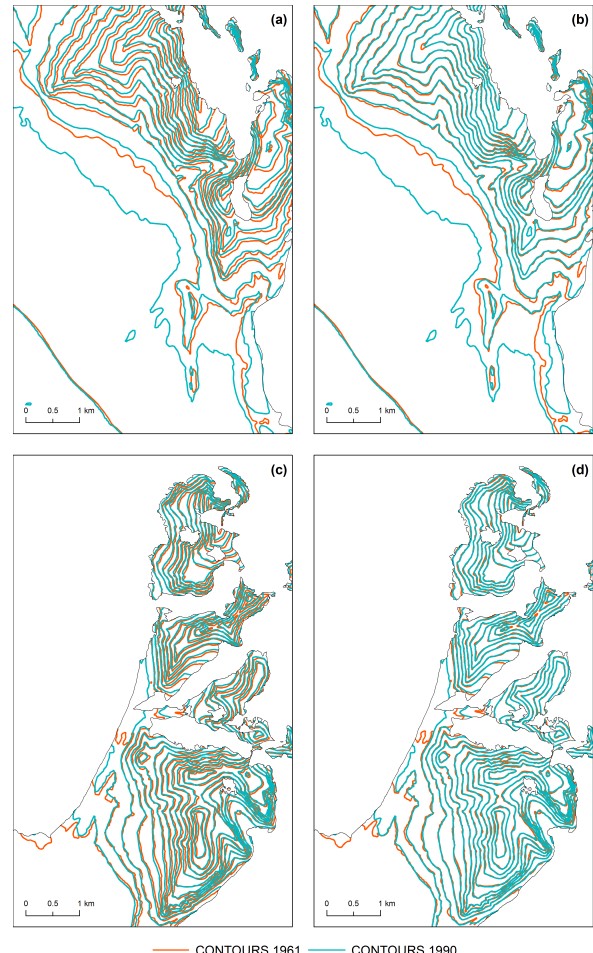

**Figure 12.** Course of contour lines in non-glaciated areas in the western part of sheet 10 – Bungebreen: georeference based on: (a, c) nodes of the cartographic grid and (b, d) elevation points.

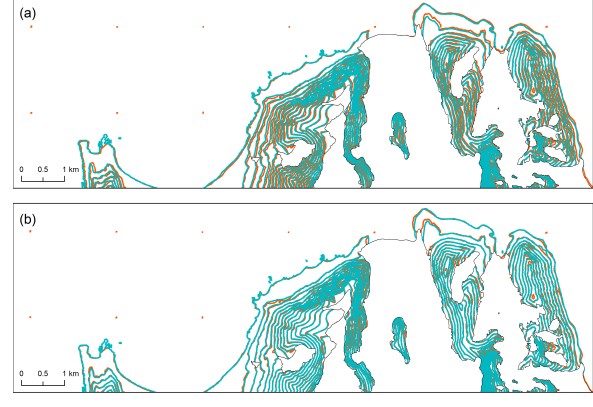

**Figure 13.** Course of contour lines in non-glaciated areas in the southern part of sheet 5 – Hornsund: georeference based on: (a) nodes of the cartographic grid; and (b) elevation points.

ing reference elevation data were evaluated. To this end, the final 1961 IGF PAN elevation model was subtracted from the 2010 NPI model, and elevation differences between the models in non-glaciated areas were analysed in individual slope classes.

To assess the usefulness of the DEM in studying changes in glacier thickness, it can be assumed that its vertical accuracy for non-glacial areas with a slope of less than 30° will also apply to the surface of most glaciers, because their slope usually falls into this class. The analysis in the following will therefore focus on such areas. Apart from the surface of steep slopes, the evaluation also excludes non-glaciated areas that cannot be considered stable because the differences in elevation between the two models may result from processes going on in the natural environment, e.g. melting of dead ice in marginal zones of glaciers, or the accumulation or erosion activity of proglacial streams in their forefields. Areas of steep or very steep slopes presented on the IGF PAN maps as a rock signature could also not be verified.

After considering the aforementioned criteria, evaluation for vertical shift was performed for each map sheet. The part of the IGF PAN model selected for the vertical error analysis covered 88.1 km$^2$, which constituted 37.3% of non-glaciated areas (236.3 km$^2$) and 14% of the entire land area (632.3 km$^2$) analysed within this model. For comparison, the area covered by glaciers was 396.0 km$^2$ (62.6% of the studied land area), and the area of steep and very steep slopes was 139.8 km$^2$ (22.1%).

Verification of the vertical error of the 1961 model began with classification of slopes by gradient. To this end, a slope map was first created and then reclassified to distinguish two slope classes for the area: 0-30° and >30°. Next, the reclassified raster was transformed to a vector layer, from which polygons of the second slope class (steep and very steep slopes) were removed, as were glacier surfaces (extent from 1961), marginal zones, extra-marginal sandurs, glacial river beds, lakes and seas. The resulting mask was used to select areas of elevation differences between the years 1961 and 2010 from the raster, and these areas were those that would be evaluated in terms of vertical accuracy. In the last step, this model was corrected by subtracting the mean difference obtained for each map sheet. After correction, all maps sheets were merged. The final mean elevation difference (bias) between the compared models on stable terrain was 0.0 m, the mean absolute error (MAE) 2.66 m, and the standard deviation of differences 2.93 m. The results of comparisons of the final 1961 model against the 2010 reference model are presented in Figure 10d.

### 4.4    1961–1990–2010 changes in glacier geometries

The measure to examine the extent and pattern of glacier retreat in the years 1961–1990–2010 was changes in their surface area, the rate of frontal recession, and, where data allowed (i.e., for land-terminating glaciers), changes in thickness. This analysis covered 28 glaciers that lay within the analysed sheets of the 1961 map. After initial classification into two glacier types (land-terminating and tidewater), changes in their geometry were calculated.

## 5    Results

In the study period, most glaciers on the mainland of northwestern Sørkapp Land were in recession, as reflected in a decrease in the total area of nearly 7.2% – from 74.8 km$^2$ in 1961 to about 69.4 km$^2$ in 2010. The average rate of change in the surface area of the region's land-terminating glaciers was 0.19 km$^2$, i.e. about 0.2% of glaciated area per year (Table 3).

The pace of surface recession on western Sørkapp Land during the study period 1961-1990 varied between individual land-terminating glaciers. In terms of surface area and ice mass loss, the recession was greatest for the largest glaciers in the region: Gåsbreen and Bungebreen. For these glaciers, the changes are most pronounced in the lower parts of their snouts (Table 3, Fig. 14).

Of the largest glaciers in the region, though in retreat, the snout of the westernmost glacier (Gåsbreen) was in 1961 still piled up on the eastern slopes of the Wurmbrandegga–Savičtoppen ridge to an elevation of 150 m a.s.l. In the period 1961–1990, the Gåsbreen's recession manifested itself primarily as a narrowing and thinning of the lowest parts of the glacier with area decrease by 1.65 km$^2$, while the frontal retreat was relatively small, amounting to about 320 m (11 m/year). Meanwhile, its frontal part was significantly lowered by up to 83 m at the line of its 1990 extent. Outside the frontal and lateral parts, the lowering of the glacier surface gradually became less intense upward, while thickening was observed in the accumulation zone. In the years 1990-2010, the decrease in glacier area continued to progress, reaching 0.96 km$^2$. The manner of the recession changed in this period, which was reflected in a significant shortening of the glacial tongue by 717 m (35.9 m/year) and its slightly less intense lowering than in the previous period, with a maximum of 57 m.

Similar patterns of change in geometry (expressed as thickness increasing in the accumulation zone and decreasing in the ablation zone, combined with a clear retreat of the terminus) were observed for the Bungebreen glacier. In the period 1961-1990, the glacier area decreased by 2.9 km$^2$, and the frontal retreat was over 1,300 m (46 m/year). The changes in glacier extent were accompanied by a severe lowering of the surface of the lower parts of the snout, of up to 85 m at the line of its extent in 1990. However, against this background, the area of medial moraine stood out, as it played a protective role and attenuated the surface lowering. Here and there, the upper parts of the glacier built up during this period. Because Bungebreen is a compound valley glacier, supplied by several firn fields, this building-up was not uniform

**Table 3.** Differences in the area of land-terminating glaciers in north-western Sørkapp Land, 1961–1990-2010.

| Glacier | Area (km$^2$) | | | Area change (km$^2$) (%) | | | Area change rate (km$^2$/yr) (%) | | |
|---|---|---|---|---|---|---|---|---|---|
| | 1961 | 1990 | 2010 | 1961–1990 | 1990–2010 | 1961–2010 | 1961–1990 | 1990–2010 | 1961–2010 |
| Arkfjellbreen | 0.78 | 0.73 | 0.67 | -0.05 (-6.4) | -0.06 (-8.2) | -0.11 (-14.1) | -0.002 (-0.2) | -0.003 (-0.4) | -0.002 (-0.3) |
| Bautabreen | 0.84 | 0.78 | 0.61 | -0.06 (-7.1) | -0.17 (-21.8) | -0.23 (-27.4) | -0.002 (-0.3) | -0.009 (-1.1) | -0.005 (-0.6) |
| Bungebreen | 49.61 | 46.63 | 43.56 | -2.98 (- 6.0) | -3.07 (-6.6) | -6.05 (-12.2) | -0.103 (-0.2) | -0.15 (-0.3) | -0.123 (-0.3) |
| Gåsbreen | 13.99 | 12.34 | 11.38 | -1.65 (-11.8) | -0.96 (-7.8) | -2.61 (-18.7) | -0.06 (-0.4) | -0,05 (-0.4) | -0.05 (-0.4) |
| Goësbreen | 1.19 | 0.94 | 0.28 | -0.25 (-21.0) | -0.7 (-70.2) | -0.9 (-76.5) | -0.009 (-0.7) | -0.3 (-3.5) | -0.02 (-1.6) |
| Gråkallbreen | 0.16 | 0.14 | 0.03 | -0.02 (-12.5) | -0.11 (-78.6) | -0.13 (-81.3) | -0.001 (-0.4) | -0.005 (-3.9) | -0.003 (-1.7) |
| Mehestbreen | 3.08 | 3.04 | 3.01 | -0.04 (-1.3) | -0.03 (-1.0) | -0.07 (-2.3) | -0.001 (0.0) | -0.002 (-0.1) | -0.001 (0.0) |
| Mikaelbreen | 3.73 | 3.72 | 3.35 | -0.01 (-0.3) | -0.37 (-9.9) | -0.38 (-10.2) | 0.000 (0.0) | -0.019 (-0.5) | -0.008 (-0.2) |
| Nigerbreen | 0.29 | 0.26 | 0.25 | -0.03 (-10.3) | -0.01 (-3.8) | -0.04 (-13.8) | -0.001 (-0.4) | -0.001 (-0.2) | -0.001 (-0.3) |
| Nordfallbreen | 0.83 | 0.80 | 0.76 | -0.03 (-3.6) | -0.04 (-5.0) | -0.07 (-8.4) | -0.001 (-0.1) | -0.002 (-0.3) | -0,001 (-0.2) |
| Påskefjella gl. | 1.15 | 1.08 | 1.05 | -0.07 (-6.1) | -0.03 (-2.8) | -0.1 (-8.7) | -0.002 (-0.2) | -0.002 (-0.1) | -0.002 (-0.2) |
| Plogbreen | 0.76 | 0.64 | 0.60 | -0.12 (-15.8) | -0.04 (-6.3) | -0.16 (-21.1) | -0.004 (-0.5) | -0.002 (-0.3) | -0.003 (-0.4) |
| Portbreen | 0.56 | 0.51 | 0.34 | -0.05 (-8.93) | -0.17 (-33.33) | -0.22 (-39.3) | -0.002 (-0.3) | -0.009 (-1.7) | -0.004 (-0.8) |
| Reischachbr. | 0.35 | 0.31 | 0.25 | -0.04 (-11.43) | -0.06 (-19.35) | -0.1 (-28.6) | -0.001 (-0.4) | -0.003 (-1.0) | -0.002 (-0.6) |
| Signybreen | 3.33 | 2.45 | 1.94 | -0.88 (-26.43) | -0.51 (-20.82) | -1.39 (-41.7) | -0.030 (-0.9) | -0.026 (-1.0) | -0.028 (-0.9) |
| Silesiabreen | 0.24 | 0.22 | 0.20 | -0.02 (-8.33) | -0.02 (-9.09) | -0.04 (-16.7) | -0.001 (-0.3) | -0.001 (-0.5) | -0.001 (-0.3) |
| Smaleggbreen | 1.94 | 1.42 | 1.08 | -0.52 (-26.80) | -0.34 (-23.94) | -0.86 (-44.3) | -0.018 (-0.9) | -0.017 (-1.2) | -0.018 (-0.9) |
| Sokolovbreen | 0.96 | 0.92 | 0.85 | -0.04 (-4.17) | -0.07 (-7.61) | -0.11 (-11.5) | -0.001 (-0.1) | -0.004 (-0.4) | -0.002 (-0.2) |
| Svalisbreen tr. | 1.68 | 1.37 | 0.96 | -0.31 (-18.45) | -0.41 (-29.93) | -0.72 (-42.9) | -0.011 (-0.6) | -0.021 (-1.5) | -0.015 (-0.9) |
| Wiederbreen | 2.03 | 1.87 | 1.73 | -0.16 (-7.88) | -0.14 (-7.49) | -0.3 (-14.8) | -0.006 (-0.3) | -0.007 (-0.4) | -0.006 (-0.3) |
| Total | 87.5 | 80.2 | 72.9 | -7.3 (8.4) | -7.3 (9.1) | 14.6 (16.7) | | | |

throughout the accumulation zone. An increase in glacier thickness of up to 20 m was recorded primarily in parts with a favourable topographic setting, i.e., where ablation is limited by a northern exposure or by being shaded by the steep slopes of the Gråkallen, Kalksteinstupa and Stupryggen massifs. There was also an approximately 10 m increase in thickness in the ice flowing northward from the Kvitgubben and Lysentoppen massifs. On the contrary, zero or slightly negative values were recorded in the upper southerly-exposed parts of the ice-filled passes on Hestskankfallet and Vasil'evskaret, although there was also a small area of increased thickness here (Fig. 14a). The accumulation of a large mass of snow in the upper part of the glacier, with a simultaneous decrease in thickness in its lower parts, leads to the fact that it becomes steeper, resulting in a slow increase in stress and, consequently, acceleration of movement. In 2007, when the thickness of the ice layer in the upper part of Bungebreen reached a critical value, a glacier surge was triggered (Sund et al. 2009). The mass of ice accumulated over the years moved downward as a kinematic wave, causing large changes in the geometry of the glacier. Rapid drainage of ice from the reservoir zone led to its reduction in several places to about 30-31 m, while in the lower parts there was an increase in the thickness of the glacial tongue by about 22 m. The acceleration of the glacier's movement was accompanied by its advance. In 2007, Bungebreen's front moved 112 m. This process con-

tinued in the following years, reaching an additional 187 m in 2007-2010.

In the years 1961–2010, a very large percentage of area loss was also observed in the western and low-lying small-valley Gråkallbreen, Goësbreen and Portbreen glaciers. This process was accompanied by significant thinning, often along the longitudinal profile, and totalled 20-40 m in the upper parts and 40-60 m at their termini. The decrease in thickness was very clearly marked in these glaciers, especially in the central and lower parts, which in the case of the Portbreen glacier, for example, led to the ice cover partially disappearing and fragmenting into smaller ice lobes separated by a rock step (Fig. 14c).

Against the backdrop of glaciers that have undergone significant changes over the analysed decades (seen mainly in a significant loss of ice mass), two glaciers stand out as having undergone relatively little change in geometry. These are the Nordfallbreen and Mehestbreen glaciers. Between 1961 and 2010, the Nordfallbreen area decreased by only 0.07 km$^2$, i.e. 8.4%, which is among the lowest values in the entire region (Table 3). The extent of the glacier went virtually unchanged in 1961-1990. Only after 1990 did Nordfallbreen slightly retreat, losing 38 m of its length. However, the slight changes in surface area and extent were accompanied by a thinning. This was less than in other glaciers in the region and ranged from 20–30 m in the ablation zone to 8–13 m in the accumulation zone (Fig. 14). Even smaller changes in

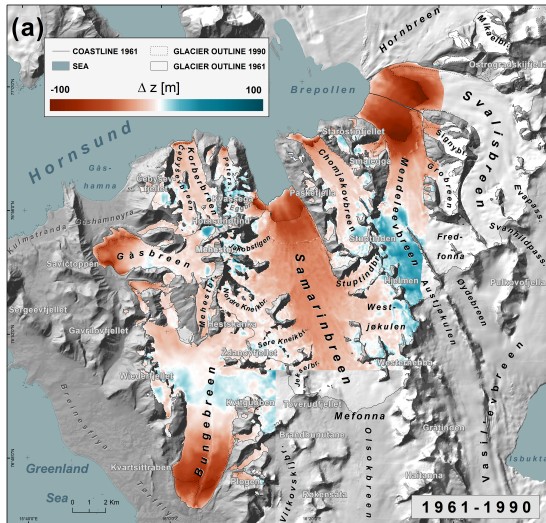

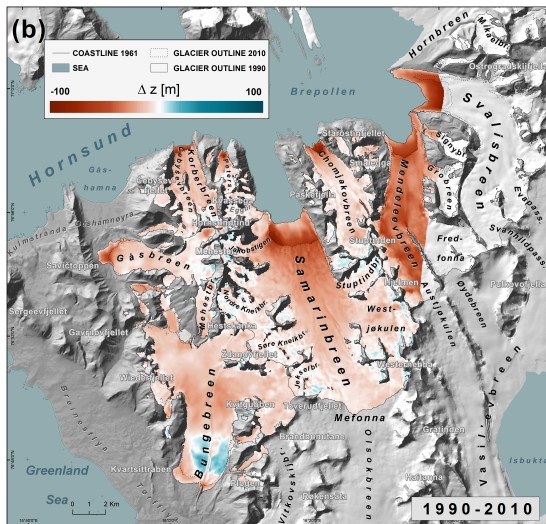

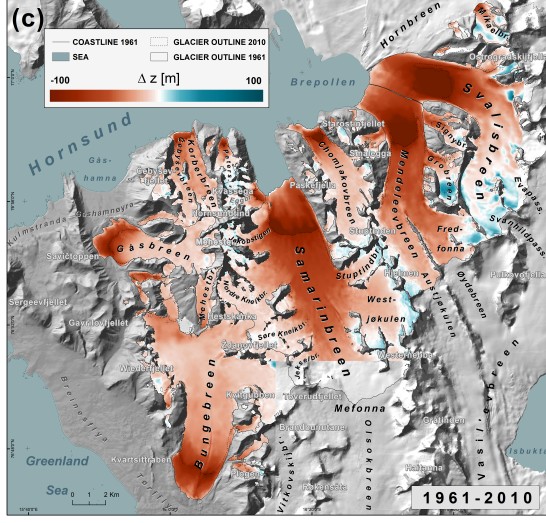

**Figure 14.** Glacier elevation change in the northwestern Sørkapp Land, 1961–1990-2010: (a) 1961-1990, (b) 1990-2010, (c) 1961-2010.

geometry were recorded for the Mehestbreen. During the entire study period, its area decreased only by 0.07 km² (i.e., 2.3%) and the glacier terminus receded by 120 m in 1961-1990, while in the next research period, 1990-2010, it was only about 10 m (Table 3). Analysis of elevation differences in the glacier longitudinal profile reveals that thinning in the years 1961-1990 was greatest in the lower parts of the ablation zone, at 30-40 m, while the accumulation zone actually increased in thickness by about 10–15 m (Fig. 14c).

In the north of the area, which is dominated by glaciers flowing into the Hornsund Fjord, the disappearance of the ice cover was mainly the result of iceberg calving. The surface area of the eight analysed calving glaciers fell from 210 km² in 1961 to 177 km² in 2010, constituting a 15.9% decrease (Table 4). The average recession rate of the calving glaciers in 1961–2010 was 0.01% per year. For the region's largest glaciers on the west, which flow directly into the Hornsund fjord (i.e. Körberbreen and Petersbreen), the areal decrease was 7-9%, while for the largest glaciers further east it was much larger 22-26%. Smaller glaciers calving into Samarinvågen Bay, lost from 13% in Kvasseggbreen to 16.6% in Eggbreen (Table 3).

Changes in surface area were accompanied by changes in ice thickness. In their ablation zones, which are subject to greater insolation, the thickness decreased and a general frontal retreat was observed. This differed in size and pace for individual glaciers (Fig. 14, Table 4 and 5). In the case of the Körberbreen glacier, the maximum lowering of the frontal parts (between the 1984 and 2010 extents) did not exceed 100 m. However, further eastward in the former tributaries of the Samarinbreen and in Petersbreen this lowering was greater, reaching 105 m for the Kvasseggbreen snout, 120 m in parts of the Eggbreen, and up to as much as 125 m in Peterbreen. In the higher parts of the glaciers studied, there was a clear build-up of firn fields in this period. In the east, in the largest calving glaciers of the study area, Mendele'evbreen and Svalisbreen, this lowering reached as much as 120 m (Fig. 14).

## 6  Discussion

The accuracy of simulations prognosing changes in glacier volumes based on dynamics models depends largely on that those models have been initialised correctly (Oerlemans, 1997; Collao-Barrios et al., 2018). Glaciers differ in response time to changes in mass balance, and this requires that data on the geometry of glaciers should go back as far as possible – preferably to a state of equilibrium with climatic conditions (Zekollari et al., 2020). If this is not possible, these models can be properly calibrated and verified using later data; nevertheless, the further back the data go, the better and the more accurately future changes can be predicted. Thus, any glacier topography data from the 1960s are extremely valuable (Andreassen et al., 2020). There is little data available for the

**Table 4.** Differences in the area of tidewater glaciers in north-western Sørkapp Land, 1961–1990-2010.

| Glacier | Area km² | | | Area change (km²) (%) | | | Area change rate (km²/yr) (%) | | |
|---|---|---|---|---|---|---|---|---|---|
| | 1961 | 1990 | 2010 | 1961–1990 | 1990–2010 | 1961–2010 | 1961–1990 | 1990–2010 | 1961–2010 |
| Körber | 10.8 | 10.5 | 10.0 | -0.25 (-2.32) | -0.55 (-5.22) | -0.80 (-7.41) | -0.01 (-0.08) | -0.03 (-0.26) | -0.02 (-0.002) |
| Peters | 2.3 | 2.2 | 2.1 | -0.07 (-3.03) | -0.12 (-5.36) | -0.19 (-8.23) | -0,002 (-0.10) | -0.01 (-0.27) | -0,004 (-0,002) |
| Kvassegg | 0.9 | 0.8 | 0.8 | -0.09 (-10.11) | -0.03 (-3.75) | -0.12 (-13.48) | -0,003 (-0.35) | -0,002 (-0.19) | -0,002 (-0.01) |
| Egg | 2.3 | 1.9 | 1.9 | -0.35 (-15.28) | -0.03 (-1.55) | -0.38 (-16.59) | -0.01 (-0.53) | -0,002 (-0.08) | -0.01 (-0.01) |
| Samarin | 86.6 | 82.9 | 78.5 | -3.32 (-3.85) | -4.47 (-5.39) | -7.79 (-9.03) | -0.11 (-0.13) | -0.22 (-0.27) | -0.16 (-0.003) |
| Chomjakov | 15.3 | 14.5 | 14.0 | -0.83 (-5.41) | -0.52 (-3.59) | -1.35 (-8.81) | -0.03 (-0.19) | -0.03 (-0.18) | -0.03 (-0.004) |
| Mendeleev | 45.2 | 38.5 | 35.0 | -6.67 (-14.77) | -3.50 (-9.10) | -10.17 (-22.52) | -0.23 (-0.51) | -0.18 (-0.45) | -0.21 (-0.01) |
| Svalis | 47.0 | 41.4 | 34.5 | -5.58 (-11.87) | -6.96 (-16.81) | -12.54 (-26.69) | -0.19 (-0.41) | -0.35 (-0.84) | -0.26 (-0.01) |
| Total | 210.0 | 192.8 | 176.7 | -17.16 (-8.17) | -16.18 (-8.39) | -33.34 (-15.88) | -0.59 (-0.28) | -0.81 (-0.42) | -0.68 (-0.01) |

**Table 5.** Surface elevation change of the glaciers in north-western Sørkapp Land, 1961–2010.

| Glacier | Mean elevation change rate (m/yr) |
|---|---|
| Arkfjellbreen | -0.40±0.05 |
| Bautabreen | -0.16±0.05 |
| Bungebreen | -0.51±0.05 |
| Chomjakovbreen | -0.39±0.05 |
| Eggbreen | -0.42±0.05 |
| Gåsbreen | -0.17±0.05 |
| Goësbreen | -0.68±0.05 |
| Gråkallbreen | -0.44±0.05 |
| Körberbreen | -0.42±0.05 |
| Kvassegggbreen | -0.11±0.05 |
| Mehestbreen | -0.36±0.05 |
| Mendeleevbreen | -0.71±0.05 |
| Mikaelbreen | -0.17±0.05 |
| Nigerbreen | -0.20±0.05 |
| Nordfallbreen | -0.38±0.05 |
| Påskefjella gl. | -0.06±0.05 |
| Petersbreen | -0.35±0.05 |
| Plogbreen | -0.35±0.05 |
| Portbreen | -0.41±0.05 |
| Reischachbr. | 0.35±0.05 |
| Samarinbreen | -0.62±0.05 |
| Signybreen | -0.40±0.05 |
| Silesiabreen | -0.17±0.05 |
| Smaleggbreen | -0.36±0.05 |
| Sokolovbreen | -0.46±0.05 |
| Svalisbreen | -0.67±0.05 |
| Svalisbreen tr. | -0.26±0.05 |
| Wiederbreen | -0.23±0.05 |
| All glaciers | -0.56±0.05 |

Svalbard region in this period, highlighting the importance of the results presented here.

The disappearance of ice in the western Sørkapp Land in 1961–2010 was the result of various processes. It was caused by both surface melting of ice and the breaking-off of ice-bergs during calving. Both processes had a significant impact on the overall mass loss from the glaciers on Sørkapp Land. They are estimated to be responsible for 79% and 21%, respectively, of the overall mass loss from glaciers across Svalbard (Błaszczyk et al., 2009).

Important factors influencing the ablation of glaciers flowing into Hornsund Fjord in the western part of the Sørkapp Land peninsula are the northern and eastern exposures of their accumulation zones and the significant shading of their surfaces by high mountain ranges. For this reason, the winter snow cover here lasts longer and is thicker, and the ablation is weaker relative to neighbouring glaciers with western exposures (Jania, 1987). Higher accumulation and some reduction in glacier ablation also result from their accumulation zones reaching more than 700 m a.s.l. and being surrounded by the steep slopes of the massifs that supply them with additional snow (Jania, 1987).

It can be seen that the interaction of all these factors has clearly increased the thickness of firn and ice in the highest and middle parts of the glaciers that flow into the Hornsund fjord during the years 1961–2010 (Fig. 14). At the same time, the changes in position of the thickened parts of the Körberbreen and Petersbreen glaciers are noteworthy, as shown by studies of changes in the range and speed of the Körberbreen in shorter time intervals (Pillewizer, 1939; Jania, 1987; Ziaja and Dudek, 2011; Błaszczyk et al., 2013). This suggests, in line with the supposition of Jania (1987), regular short-term displacement of the kinematic waves of ice that are characteristic of surging (especially in relation to the Körberbreen glacier). The research period adopted here (on the order of several decades) is too long to properly detect and illustrate this phenomenon, but other studies for this area provide evidence supporting the thesis.

On land, glacial systems evolved at variable rates, which can be associated with variable topoclimatic and local conditions in the western Sørkapp Land. The recession was fastest in the western and southernmost glaciers of the region, where the air masses of the Greenland Sea and the warm West

Spitsbergen current are in effect (Ziaja et al., 2016). Aside from clear frontal retreat, there was also a significant decrease in thickness in their longitudinal profiles. In the small, westward, low-lying valley glaciers this was especially pronounced, especially in the middle and lower parts of the snouts, where smaller patches of dead ice emerged.

Although glacial recession was the predominant phenomenon in land-terminating glaciers in the western Sørkapp Land, the warming effect was mitigated in some places by the terrain and the significant elevation of the mountain massifs from which some of the glaciers originate. Being favourably located either at a significant elevation or in the shadow of high mountains stabilised the situation somewhat for some glaciers here, because their maintenance or local increase of mass was favoured by both an orographic increase in snowfall and additional supply from avalanches. This applies, for example, to Nordfallbreen, which is shaded from the south, and small glaciers originating on the slopes of the Hornsundtind and Kvassegga groups of mountains.

Nordafallbreen is adjacent to Nordfallet (824 m a.s.l.) to the south, which shades it against the sun while also providing it with additional supply by avalanches. Mehestbreen is similarly fed, being bordered to the east by the Mehesten (1,383 m a.s.l.) and Hestskanka (997 m a.s.l.) massifs, and by Hoven hill (869 m a.s.l.) to the north. Their influence is seen in the spatial distribution of positive values on the glacier elevation change maps, more of which lie closer to the eastern edge of the glacier. An additional factor limiting ablation on the Mehestbreen is its significant elevation, which puts a large part of the glacier's surface above the mass balance equilibrium line (300-400 m a.s.l.).

There are few studies that the results of this study of the peninsula's surface glaciation recession can be compared against. In the older literature, such analyses were carried out for individual glaciers (Jania, 1987; Schöner and Schöner, 1997) or at the regional scale at best (Jania, 1988a). However, the observed trends in the elevations of the Sørkapp Land glaciers in 1961–2010 are comparable to other areas of Spitsbergen, although the number of studies with similar temporal coverage is limited (Nuth et al., 2010; Małecki, 2013; Błaszczyk et al., 2013).

## 7   Conclusions

Correctly assessing the utility of the series of maps issued by the Institute of Geophysics of the Polish Academy of Sciences is very important in order to precisely determine changes in glacier geometries in the western Sørkapp Land. Although the IGF PAN field campaign was conducted in the early 1980s, the maps published after the expedition were based on elevation data taken from aerial photos from 1961, upon which only glacier extents were updated (with a change in colour of contours). Crucially, contour lines were not up-

dated in this 1984 edition and continued to represent the higher elevations of 1961.

In response to this, the map coordinates on the 1961 map have now been corrected, so that it can be used for comparative analyses of changes in glacier surface elevations over the years 1961-1990-2010. This is especially true for glaciers that are land-terminating, for which data relating to their entire surface area are now corrected and complete.

However, the value of data on tidewater glaciers for various types of comparison is limited to their upper parts (above the line of their 1984 extents). This is because updating their extents in 1984 required that contour lines between the extents designated for 1961 and 1984 be deleted and that the elevation of this surface be zeroed on the map, i.e., brought to sea level. Therefore, when analysing the IGF PAN sheets, it is impossible to determine the exact height of the ice cliffs of the Körberbreen, Petersbreen, Kvaseggbreen and Eggbreen tidewater glaciers in 1961.

Consequently, this study finds that in the years 1961–2010, the maximum lowering of surface was about 90–100 m in the largest land-terminating glaciers on the peninsula, and over 120 m in tidewater glaciers (above the line marking their 1984 extents), with mean glacier elevation change rate of -0.56±0.05 m/yr over the entire region. Glaciated areas aside, the surface-corrected IGF PAN maps can also be used to analyse landscape dynamics, including changes that occur in marginal zones.

## 8   Data availability

All data is available at Zenodo service (https://doi.org/10.5281/zenodo.4573129) (Dudek and Pętlicki, 2021). Data format: ESRI shapefile and GeoTIFF. The datasets contain vector layers (topographic and glacier outlines) and Digital Elevation Model (DEM) covering north western part of Sørkapp Land peninsula, Svalbard, for the year 1961. The shape file *glacier_1961_northwestern _Sorkappland.shp* contains the glacier areas manually delineated from vertical aerial photos captured during the historical photogrammetric overflight commissioned by the Norwegian Polar Institute on August 24 and 25, 1961. The shape file *contour_1961_10m_northwestern_Sorkappland.shp* contains contour lines with intervals of 10 m based on digitised historical maps edited in 1987 by the Institute of Geophysics of the Polish Academy of Sciences and registered using a cartographic grid and elevation points. The shape file *peak_1961_northwestern_Sorkappland.shp* contains elevation points – topographic and triangulation – used in the vector data registration process. Shape file *rock_1961_northwestern_Sorkappland.shp* delineates areas very steep presented on the source maps as rock cliff symbols. This file also indicates areas where the highest elevation errors in the generated Digital Elevation Model are plausible. All shape files were produced in the UTM

projection system (northern hemisphere, zone 33) based on the ETRS89 ellipsoid (datum D_ETRS_1989). The raster file *dem_1961_5m_northwestern_Sorkappland.tif* contains Digital Elevation Model (DEM) with 5 m resolution generated from corrected contour lines.

**Author contributions.** JD conceived the study, processed and analysed the data, drafted the manuscript. MP contributed to the discussion, review and editing of the manuscript.

**Competing interests.** No competing interests are present

**Acknowledgements.** This work was funded by the European Economic Area Financial Mechanism and the Norwegian Financial Mechanism (EEA and Norway Grants), Svalbard Science Forum (SSF) Arctic Field Grant 2016 (RIS-ID 10414) and by a grant from the Priority Research Area (Anthropocene) under the Strategic Programme Excellence Initiative at Jagiellonian University. JD would like to thank prof. Jon Ove Hagen for enabling her research stay at the University of Oslo, Tim Brombley for editorial support, Harald Faste Aas for the information regarding NPI surveys and data, Bartłomiej Luks for recovering the original unfolded paper version of all map sheets from the archives of IGF PAN, and Alexandre Bevington, Mateusz Suwiński and Pamela Dobiesz for their help with digitisation of map sheet 3 - Hornbreen, 5 - Hornsund, 6 - Brepollen, and 10 - Bungebreen. We would like to thank our Handling Editor Reinhard Drews and three anonymous reviewers for their significant input in the revised version of the manuscript.

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
