# Peer review of "Unlocking archival maps of the Hornsund fjord area for monitoring glaciers of the Sørkapp Land peninsula, Svalbard"

_Earth System Science Data, 2021_

## Author Comment (AC1)

**General comments:**

R: Dudek & Pętlicki digitize and georeference three topographic map sheets covering the western portion of Sørkapp Land, Svalbard. They clearly outline the steps they took to digitize the contour lines, generate a raster digital elevation model (DEM), use triangulation points to co-register the historical map to a more modern (1990) DEM of the area, and assess the uncertainty in the final historical DEM using elevation differences to the reference DEM over non-glaciated terrain. A notable contribution of this paper is that it establishes that the contour lines in the IGF PAN map sheets represent the 1961 glacier elevations, rather than the elevations from the 1980s, which is when the maps were updated and published.

A: We appreciate your valuable feedback and we corrected our manuscript (the text and figures) according to these suggestions. We believe that they helped substantially to improve the quality of our work. Below we included our answers. Referee text (R) and author responses (A) are indicated.

R: In my view, there are two categories of shortcomings in this work. The first pertains to the question of whether this paper presents a substantial new dataset to the community—a question that the Topic Editor raised some skepticism about. In lines 97-100, the authors explain that they digitized three of the ten sheets in the IGF PAN topographic map. But they don't provide an explanation for why they didn't digitize the other seven. Since Dudek & Pętlicki reconstruct the historical (1961) geometry of just a handful of the ~1,600 glaciers in Svalbard (König et al., 2014), this manuscript reads more like a nice recipe for digitizing historical topographic maps for analysis of geodetic glacier change, rather than presenting a large new dataset.

A: In our manuscript, we focused on the map sheets of the PAS series that in our opinion presented the highest value for glaciological research. The three sheets we initially have chosen for processing (5, 8, 10) cover the peninsula further inland and present the complete surface of 14 land-based glaciers. We agree that this is only a small sample of glaciers of Svalbard, nonetheless a very important one. The remaining map sheets were already partly covered by the DEM published by NPI (based on original aerial photographs from 1961), or presented only frontal parts of the tidewater glacier (updated to the period of 1982-1984, when PAS field measurements took place), or did not cover the glaciers. We still believe that our initial choice was valid, nonetheless, we also agree that it would be better to show larger spatial coverage. For the revised version of our manuscript, we decided to significantly enlarge our study both spatially and temporally by adding the 2010 data and additional 4 map sheets (as shown in the attached figure). This way we will use all 7 maps produced for Sørkapp Land and overlapping with the data from the latest flight campaign.

[Figure]

1 - Werenskioldbreen
2 - Burgerbukta
3 - Hornbreen
4 - Isbjornhamna
5 - Hornsund
6 - Brepollen
7 - Pallfyodden
8 - Gasbreen
9 - Samarinbreen
10 - Bungebreen

**Legend**

CONTOUR LINES
INVESTIGATED GLACIERS
DEM NPI 1961
DEM NPI 2010

R: If this paper is going to be a template to help others digitize and georeference historical topographic maps, it becomes especially important that the methodology in the paper is careful and robust. That brings me to the second potential shortcoming of this work. In my view, there are three ways that the others could generate improved 1961 elevation reconstructions from the data already available to them. First, the authors use a TIN interpolation of the contour lines to create a DEM directly from the contour map. They could get more accurate glacier change observations by differencing just the contour lines to the reference DEM, and then interpolating the difference map. The reason is that glacier elevation change between two time points (dh) varies in space more smoothly than the topography at any one time point. This kind of problem is discussed tangentially in McNabb et al. (2019).

A: We agree that the applied method of georeferencing must be improved in order to reassure that the results are robust and reproducible. We did our best to add new processing steps, suggested by the reviewer, that allow data processing less dependent on the operator. Yes, we see that the glacier elevation change can be better quantified with the use of contour lines only and it will help to minimise the impact of DEM artifacts on ice elevation change estimates. We will address this in more detail and add relevant references, including Rolstad and others (2009).

R: Second, the final co-registered map (Fig. 14) still has large regions of positive dz and large regions of negative dz over ice-free land areas. Figure 14 suggests that the authors could still improve the georeferencing of the map by correcting for regional warping (see comment #3 below for some ideas how to do this).

A: Agree. We have improved our georeferencing following your suggestions, please see further responses for more details.

R: In summary, although I think that the digitization of historical datasets is an important and worthwhile task, the small coverage of the digitized maps presented here, and the fact that the digitization could be done in better ways given the methods already described in published literature, makes me question whether the manuscript warrants publication in ESSD. However, I think that digitizing more of the map sheets, and following some or all of the 3 suggestions mentioned in the paragraph above, would substantially improve this contribution.

A: We fully agree with this comment and we made our best effort to expand our study by digitizing 4 more maps and improving methods of dem processing. In the revised version we included all maps covering Sørkapp Land and overlapping with dataset from 2010.

**Specific comments:**

R: 1. If I recall correctly, the 1990 NP DEM for Sørkapp Land has a spatial resolution of 20 m. That relatively low resolution means that the DEM elevation at the pixel location of mountain peaks will consistently underestimate the peak elevation. How do your georeferencing results compare when you only georeference to steep mountain peaks vs. only georeference to flatter terrain?

Also, now that Norsk Polarinstitutt (NP) has released a 5 m regional DEM (2010) for Sørkapp Land (geodata.npolar.no), you might want to experiment with the georeferencing results when you use the 5 m reference. Finally, for your discussion, you might consider evaluating the two time periods (1961-1990 vs. 1990-2010). Were thinning rates and retreat rates similar over those two periods? Have they increased/decreased?

A: Good suggestion. Now we use as a reference the 2010 NPI DEM that has a higher resolution of 5 m. We have expanded the discussion with evaluation of the ice elevation change over this two epochs. Answering the question shortly: the thinning/ thickening rate of investigated glaciers changed in the period of 1990-2010. Most of the glaciers retreated faster in the second period, however for 2 glaciers (Bungebreen, Mendeleevbreen) we observed thickening in the ablation zone and thinning in their accumulation zone – which is indicative of surging process. We also observed thickening of small glaciers lying in the highest mountain massifs. In our revised manuscript we discuss that in more detail.

R: 2. You say in lines 135-139 that you analyze the original 1961 aerial photographs from the Norwegian Polar Institute. Can you use structure-from-motion to make a 3D model from those images? This has been done successfully for Svalbard glaciers numerous times. See, for example, Mertes et al. (2017), Midgley & Tonkin (2017), Girod et al. (2018), Kavan (2020), Holmlund (2021).

A: Unfortunately, we do not have access to high quality scans of aerial imagery from 1961 that could be used to generate new DEMs using stereophotogrammetry or structure-from-motion methods. We tried to process low quality scans that we possess with Agisoft Photoscan, however the results were not satisfactory. The main focus of the study was a use and recovery of historical maps and not to generate completely new dataset from the source data, to which we do not have access.

R: 3. You do the co-registration based on the few dozen "triangulation points" listed in Tables 2-4. How would the co-registration differ if you instead applied the Nuth & Kaab (2011) procedure using all DEM pixels in stable ice-free areas? The reason I ask is that, in Figure 14, there remains a considerable amount of spatial structure in the dz map over land areas (i.e., large regions that are consistently blue, transitioning to large regions that are consistently red). That suggests that you might want to change the way you do the co-registration. For example, you could compute the dx, dy, dz offsets using the Nuth & Kaab (2011) method for individual patches of ~2 km x ~2 km (and you should explore the sensitivity to that window size), then unwarp the map using the vector field you generate, where the "vector field" is the <dx, dy, dz> vectors at the grid of locations where you extracted co-registration chips).

A: We applied original Nuth & Kaab (2011) method of DEM co-registering but it did not help much as still the models exhibited some warping. Following this comment of the reviewer we

applied the co-registration method in a moving window and it indeed improved the results. The window size has to be much larger though (approx. 5x5km) because the eastern Sorkapp Land is highly glaciated and there are very little rock outcrops that can be used as a reference for co-registration. The drawback of this method is that it does not make use of available point features from the map such as mountain top positions and therefore we are inclined to use it as a second step of data processing, after the rubber-band correction of the isolines.

**Line comments:**

R: Line 2: you might consider changing "reliable comparative" to "quantitative"
A: corrected

R: Line 9: change "from" to "on"
A: corrected

R: Lines 13-15: this first sentence needs a citation. Also, the word "dynamic" connotes ice dynamics (e.g., the flow of ice, surge behavior, etc.). But it seems like you are trying to say that melting is happening more rapidly in Svalbard than other places in the world. Perhaps you could start by saying that Svalbard is warming more rapidly than elsewhere (Nordli 2020), and then relate that warming trend to negative mass balance (e.g., Nuth et al., 2010; Morris et al., 2020). Or, you could skip straight to the negative mass balance.
A: By 'more dynamic' we mean that changes in Spitsbergen (such as progressive disappearance of ice) are occurring faster than in other parts of the European Arctic (and the world). We will rephrase this sentence and add suggested references.

R: Line 26: What is "this scientific field" referring to? Glaciology or remote sensing?
A: Glaciology. In the revised manuscript we rephrased this sentence to make it clear.

R: Lines 54-55: Clarify what you mean about aerial images being "more competitive" than terrestrial photographs, and why.
A: We will rephrase this sentence. By 'more competitive' we meant more useful for the research covering larger area.

R: Table 1: Should you clarify in the caption that these mapping campaigns were done by the Norwegian Polar Institute, since, in the main text, you just discussed the Polish mapping efforts?
A: Corrected. We added: "Norwegian Polar Institute photogrammetric campaigns carried out over the Sørkapp Land peninsula."

R: Table 1 (a more general comment): I wonder if the information in this table would be conveyed more effectively by having a series of 8 simple line maps of Sørkapp Land, lined up side by side and colored to show the regions covered by each photogrammetric mapping campaign?
A: We would like to keep the table (with added column with references as suggested in the second review) but we also prepared and added the figure with all photogrammetric campaigns.

R: Line 60: replace "first decade of the 21st century" with "2000s"
A: Corrected.

R: Lines 65-74: These two paragraphs would benefit from a little more clarity. Is the IGF PAN topographic map for Sørkapp Land the result of analyzing the Norwegian 1960/1961 photos in a stereoplanograph?

A: IGF PAN did not provide such detailed information on the maps.

R: Line 76: Missing a space at the end of the sentence.

A: Corrected.

R: Lines 76-77: Perhaps you should cite Ziaja & Ostafin (2015) here? Is it correct to call Sørkapp Land an island if it is still connected to Spitsbergen via the isthmus?

A: In this context by the Island we mean Spitsbergen (which is the biggest Island of Svalbard archipelago). The Sørkapp Land is its southern peninsula. We rephrased that section: "Sørkapp Land is the southern  peninsula of Spitsbergen, the largest island of  the Svalbard archipelago. It is separated from the rest of Spitsbergen by a narrow glaciated isthmus of Hornbreen-Hambergbreen (Ziaja and Ostafin, 2015) and there is an ongoing speculation whether it will form a separate island when the ice is gone (*Pälli* et al., 2003; Grabiec et al, 2017)"

R: Line 79: why do you say "as many as 14 land-terminating glaciers," rather than simply "14 land-terminating glaciers"?

A: Corrected.

R: Line 97: clarify which topographic map.

A: Corrected.

R: Lines 97-100: where are the other 7 sheets? Is there a reason you chose not to digitize those ones?

A: For the revised version of this article we added another 4 sheets covering Sorkapp Land. Sheet no 7 (which does not cover any glaciers), and sheets no 3, 6, 9, covering parts of the glaciers ending in the Hornsund fjord. The fronts of tidewater glaciers (within the extents 61-84) were removed from these sheets, and this limits their suitability for glaciological research. This was the reason why primarily we focused our research on the areas with the elevation dataset covering entire glacier surfaces.

R: Line 103: cite Fig. 2 again at the end of this sentence for clarity.

A: corrected

R: Line 112: what does "made in desk research" mean?

A: The term "desk research" used in our manuscript was directly cited from the map description meaning compilation, analysis, and processing of data and information from existing sources (in this case aerial photographs).We might rephrase this sentence in the revised manuscript.

R: Line 121-122: tell the reader here what the answer is, rather than saying "the question was answered." Do the contour lines represent the glacier elevations in 1960s or the 1980s? You get to that in lines 131-134, but you might improve clarity by telling the reader up front that the glacier contour lines represent the 1961 surface, and then go into the paragraph of how you determined that (lines 123-130).

A: We will rephrase this sentence.

R: Lines 141-144: I suppose you began this research before NPI had released the 2010 DEM for Sørkapp Land. Would that be a better reference dataset, since it is 5 m resolution rather than 20 m resolution?

A: Yes, we began our work before this recent data release. In our revised manuscript we included 2010 DEM as a reference dataset and compared it to both DEMs: from 1961 and 1990.

R: Figure 4: Readers will see that this dz map resembles a hillshade map, due to delta-x and delta-y offsets between the two DEM datasets (i.e., poor co-registration--Nuth & Kaab, 2011). You should explain that in the caption. You already explain it nicely in the main text (lines 174-176), so just add a brief explanation in the caption, too.

A: Corrected.

R: Lines 172-173: You don't need to say how you subtracted one raster layer from another (i.e., which GIS module you used).

A: Corrected.

R: Figure 5: Reference Figure 2 so people know where sheet 8 comes from. Also, in Figure 2, you might consider adding (a), (b), and (c) and clearly labeling which sheet is which.

A: Corrected.

R: Lines 177-179: You don't need to tell the reader which of the 3 sheets you worked on first.

A: Corrected.

R: Lines 180: Add "elevation points" after "195"

A: Corrected.

R: In Figures 5, 6, and 8 (the maps showing triangulation points for the 3 sheets), it would be helpful if you plotted vector arrows pointing in the direction of the dx, dy offset to the NPI reference map. That would let readers see if there are consistent patterns of warping across the map sheets.

A: Good suggestion, we will add vector arrows to all figures showing triangulation points.

R: Lines 214-216: Rather than referring to "visual assessment" of the model accuracy, can you give a quantitative metric of the elevation accuracy. For example, the root mean square error (RMSE) between the 2 DEMs on ice-free land?

A: This information will be provided for all 7 map sheets in the revised manuscript, along with a global value and a related histogram.

R: Lines 224-226: This sentence will be more compelling if you provide some stats. For example, use the NPI DEM to say XX% of the glaciated area in Sørkapp Land has slopes < 20 degrees."

A: This information will be provided for all 7 map sheets in the revised manuscript

R: Line 230: remove "for obvious reasons"

A: Corrected

R: Line 242: to improve clarity, you could say "The mean elevation difference (the bias) between…"

A: Corrected

R: Line 291: Missing space after "zone"
A: Corrected

Line 308: You might want to cite Nuth et al. (2013) here, since it investigates similar datasets across Svalbard.
A: Corrected.

Line 313: Be specific: say "structure-from-motion (SfM) photogrammetry" or structure-from-motion (SfM)-multi-view stereo (MVS)" rather than "modern methods"
A: Corrected.

R: Line 341: remove "very"
A: Corrected.

R: Line 343: what does it mean for an air mass "to be in effect"?
A: This was indeed poorly translated. Here we mean that the climate of Southern Spitsbergen, controlled by the latitude, is modified by a significant thermal difference between sea masses. Warmer Atlantic water (West Spitsbergen Current, the last branch of the Gulf Stream) that reach Sørkapp Land flowing along its western coast, affects local climate conditions. This produces relatively little glaciation and more intense ablation of the small, westward, low-lying valley glaciers.

R: Lines 367-369: Rephrase this sentence for clarity and remove the word "ignorance." Make it clear that you are saying that, even though the map sheet is labeled as 1984, the glacier contour lines reflect the 1961 elevations.
A: We rephrased that sentence according to the suggestion from the second review: "Although the IGF PAN field campaign was conducted in the early 1980s, the maps published after the expedition were based on elevation data taken from aerial photos from 1961, upon which only glacier extents were updated (with a change in colour of contours). Crucially, contour lines were not updated in this 1984 edition, and continued to represent the elevations of 1961."

R: Lines 384-386: You could remove this final sentence of the manuscript.
A: Corrected.

---

## Author Comment (AC2)

**General comments:**

R: This manuscript presents a method for how to digitize and georeference archival maps, and finally evaluates their potential for quantifying changes in glacier geometry on Sørkapp Land, Svalbard. For this exercise, three topographic map sheets from the Institute of Geophysics of the Polish Academy of Sciences (IGF PAN) published in 1987 were used together with a reference dataset from 1990. The 1987 map sheets have contour lines based on aerial photos from 1961 (from the Norwegian Polar Institute; NPI), and the 1990 dataset consists of a 20-m-resolution digital elevation model (DEM) and a glacier outline vector layer (also NPI).

Overall, this is a nice study with clearly described methods, walking us through the different steps of source data processing, verification, and data fitting, as well as presents the final DEM and examines the glacier elevation and areal change. However, there are some issues that need to be addressed, mainly considering the dataset size and the potential for improving the quality of the digitization/georeferencing. I have listed my main concerns and suggestions for improvements in the specific comments below. If these questions are resolved, I think the paper could be a good contribution to the journal.

A: We appreciate your valuable feedback and we corrected our manuscript (the text and figures) according to these suggestions. We believe that they helped substantially to improve the quality of our work. Below we included our answers. Referee text (R) and author responses (A) are indicated.

**Specific comments**

R: 1. Why did you not digitize all ten IGF PAN map sheets? Digitizing these map sheets is a valuable contribution, but it would be most useful to have all of them compiled. Especially considering the initial confusion with the contour lines (i.e., that they represented the 1961 elevations, without being corrected based on field observation before publication in the 1980's). I strongly recommend digitizing the rest of the map sheets to provide a complete updated dataset with greater areal coverage. If choosing not to, please provide an explanation for why you decided to process only three (you describe this nicely in your response to the editor), as this is not obvious to the reader. To help illustrate that these are the most valuable map sheets (for glaciological research), show all of them in Figure 2 and mark the three you present in the paper.

A: In the initial version of the manuscript we focused on the map sheets of the PAS series that in our opinion presented the highest value for glaciological research. The three sheets we initially have chosen for processing (5, 8, 10) cover the peninsula further inland and present the complete surface of 14 land-based glaciers. The remaining map sheets were already partly covered by the DEM published by NPI (based on original aerial photographs from 1961), presented only frontal parts of the tidewater glacier (updated to the period of 1982-1984, when PAS field measurements took place), or did not present the glaciers. We still believe that this choice was valid, nonetheless, we also agree that it would be better to show larger spatial coverage. For the revised version of our manuscript, we decided to expand our study both spatially and temporally by adding the 2010 data and additional 4 map sheets (as shown in the attached figure). This way we will use all 7 maps produced for Sorkapp Land and overlapping with the data from the latest flight campaign.

[Figure]

1 - Werenskioldbreen
2 - Burgerbukta
3 - Hornbreen
4 - Isbjornhamna
5 - Hornsund
6 - Brepollen
7 - Pallfyodden
8 - Gasbreen
9 - Samarinbreen
10 - Bungebreen

**Legend**

CONTOUR LINES
INVESTIGATED GLACIERS
DEM NPI 1961
DEM NPI 2010

R: 2. The method section would benefit from having a figure showing the workflow, describing all the steps. That would provide a good overview of the methodology and make it more user friendly. Also, I am a bit skeptical to whether the method is innovative enough, and whether the final map (Figure 14) is really satisfactory? There are still areas with both large positive and negative dz, suggesting that the georeferencing could still be improved. Have you considered supplementing with other methods? You refer to studies using structure-from-motion to create historical DEMs from aerial imagery (Mertes et al., 2017; Midgley et al., 2017). Why not try that, using the NPI aerial photographs from 1961, to compare to the topographic maps? Another option could be DEM production by digital stereophotogrammetry on the 1961 images. For methods, see e.g., Korsgaard et al. (2016) and references therein.

A: Following the suggestions of both reviewers we improved our methods of data processing and hence the quality of the final maps. Unfortunately, we do not have access to high quality scans of aerial imagery from 1961 that could be used to generate new DEMs using stereophotogrammetry or structure-from-motion methods. We tried to process low quality scans that we possess with Agisoft Photoscan, however the results were not satisfactory. The main focus of the study was a use and recovery of historical maps and not to generate completely new dataset from the source data, to which we do not have access.

R: Why not compare the 1961 and 1990 data to the 2010 data from NPI? The spatial resolution is higher (5 m for the 2010 DEM vs. 20 m for the 1990 DEM), and it would allow you to do a two-step comparison (1961 vs. 1990, and 1990 vs. 2010), to see if the retreat and/or thinning rates have varied between these periods, and whether they have accelerated or not. Further, you could even use the 1936 oblique photos, since from what I can tell from Table 1, the entire peninsula was covered also during that survey? Several studies compare the 1936/38 maps to the 1990 DEM (e.g., Nuth et al., 2007; Girod et al., 2018), so I strongly suggest that you compare more than two years (i.e., by adding 2010 and/or 1936), since that would add something extra to this paper.

A: Good suggestion. We decided to enlarge our study by adding the 2010 DEM (as a reference dataset) to our revised manuscript and compare it to the data from 1961 and 1990.

**Overall syntax and structure**

R: The structure is generally good, but the readability would benefit from having a native English speaker reading though this manuscript. Sometimes the word choices are not optimal, the sentence structure not correct, or the sentences too long (e.g., line 80-84). This makes it somewhat difficult to understand the message, without re-reading some of the sentences. In the introduction, the paragraphs are very short, often only two sentences each. Merge some of them to get a clearer structure and give the text a better flow. I would remove the word 'glacier' after all of the glacier names, since 'breen' in the end of all names already indicates that those are glaciers (in Norwegian). Abbreviate Norwegian Polar Institute to NPI after the first use.
A: Thank you for these suggestions. We tried to apply changes according to them in our revised version of the manuscript.

**Study area**

R: Start by introducing Svalbard, the influence of the West Spitsbergen Current, strong climate gradients etc., and how the ice masses vary between different parts of the archipelago. Then move on to the Hornsund area and the glaciers there. You could also show or describe some climate data from Hornsund, to see how the temperature/precipitation have changed from 1961 to 1990. Also introduce the concept of surging glaciers.
A: We added these changes in our manuscript.

**Source material**

R: You mix present and past tense. Decide on one (I recommend past tense) and stick to it.
A: We changed this section to the past tense.

**Line by line comments**

R: Line 3: materials -> data
A: Corrected.

R: Line 8: glaciers of -> glaciers on (this is reoccurring in several places)
A: Corrected.

R: Line 9-10: Remove sentence about dataset availability. Enough to have this information in the 'Data Availability' section
A: Corrected.

R: Line 13-15: This sentence needs to be supported by 2-3 references. Also, do you with 'more dynamic' in the first part of the sentence mean faster? Or in terms of ice dynamics? This needs to be clarified. Reference suggestions: Nordli et al. (2014), Isaksen et al. (2016), Schuler et al. (2020)
A: By 'more dynamic' we mean that changes in Spitsbergen (such as progressive disappearance of ice) are occurring faster than in other parts of the European Arctic (and the world). We will rephrase this sentence and add suggested references.

R: Line 19: huge -> crucial
A: Corrected.

R: Line 20: Give some examples of 'traditional research methods', e.g., in situ stake mass balance measurements
A: Corrected.

R: Line 23: Support this statement with a couple more references, e.g., Jacob et al. (2012) and Nuth et al. (2013)
A: Corrected.

R: Line 24: Change to 'The use of remote-sensing has a number of advantages'
A: Corrected.

R: Line 44: Change to 'the University of Wrocław were held there (Zyszkowski, 1982)'
A: Corrected.

R: Line 47: has -> have
A: Corrected.

R: Line 51-52: Change to 'including primarily the Gåsbreen area'
A: Corrected.

R: Line 57: Norwegian Polar Institute (NPI)
A: Corrected.

R: Line 59-61: Do you mean that the next map covering the entire peninsula was not published until in the first decade of the 21$^{st}$ century? Also, better to write out the year (2010) instead of 'the first decade of the 21$^{st}$ century'.
A: Corrected.

R: Line 64: 49 years later -> in 2010. Additional comment: if there are data with the same spatial extent from 2010, why don't you compare the 1961 and 1990 glacier extents to the 2010 extents as well? It would be interesting to compare the glacier areas, and see if the rate of retreat and thinning have increased or decreased etc.
A: Corrected. Following previous comments, we added dataset from 2010 as well in the revised version of the manuscript.

R: Line 68: add the spatial coverage (XXX km$^2$) of this map
A: Corrected.

R: Line 69: Enough to use the abbreviation (IGF PAN) here
A: Corrected.

R: Line 73: attempts -> aims
A: Corrected.

R: Line 73-73: Change to 'geometry of glaciers **on** the **western** Sørkapp Land peninsula'
A: Corrected.

R: Line 78-79: change to 'It **hosts** 14 land-terminating **and 4 tidewater** glaciers'
A: Corrected.

R: Line 85: Remove 's' in the end of 'Körberbreen glaciers'
A: Corrected.

R: Line 88: 'outflow glaciers' should be 'outlet glaciers'?
A: Corrected.

R: Line 93-94: Change to 'They formerly served as tributary glaciers to Samarinbreen, but as its snout receded, they split from it and today constitute separate calving glaciers'
A: Corrected

R: Line 97: The **IGF PAN** topographic map
A: Corrected

R: Line 98-100: Remove 'These were the following sheets' and put '(No. 5 –Hornsund, No. 8 – Gåsbreen, and No. 10 – Bungbreen; Fig. 2)' in brackets.
A: Corrected

R: Line 102: Change to 'The topographic map sheets presented the relief, permanent and periodic watercourses, …'
A: Corrected

R: Line 106: colour from orange **(land)** to blue **(glacier)**. Additional comment: is it blue or purple?
A: Corrected. The colour of the contour lines indicating the glaciers on the maps is dark blue.

R: Line 111-112: Explain what you mean by 'somewhat non-standard mean' – what was done and in comparison to what (the standard way)? And what do you mean by 'in desk research'?
A: We rephrased this section of the manuscript to make it more clear. The term "desk research" used in our article was directly cited from the map description meaning compilation, analysis, and processing of data and information from existing sources (in this case aerial photographs). By the 'standard way', we mean collecting field information (geodetic measurements) and using it during the aerial data processing. By 'non-standard mean' we consider using the data from the field (collected 2 decades after flight campaign) after the initial processing of the images.

R: Line 116: Change the first 'extent' to 'degree'?
A: Corrected.

R: Line 117: Is "completed in the field" a quote from the map? Otherwise rephrase.
A: Yes it is a quote from the map.

R: Line 123: Remove line break
A: Corrected.

R: Line 124-127: Change to 'These studies **were based on photos from NPI's photogrammetric overflight over the west of Sørkapp Land in the summer of 1960 (Table 1) and** resulted in a publication that included a map showing the hypsometric variation of Gåsbreen and **a** hillshade that was valid for 1960 (Schöner and Schöner, 1996).'
A: Corrected.

R: Line 127: Another -> A (since referring to other glaciers)
A: Corrected.

R: Line 136: Add reference to the NPI photos
A: These photos were not published before. We obtained the information by pers. comm. with Harald Faste Aas from NPI over email.

R: Line 142: Add reference to the NPI data after first sentence.
A: Corrected.

R: Line 144: 'slightly less' – Can this be quantified?
A: This is the information provided by the NPI. We didn't process this dataset.

R: Line 145: Divide into 4 Methods, and 4.1 Source data processing and evaluation of output data accuracy
A: Corrected.

R: Line 146: stages -> steps
A: Corrected.

R: Line 153: Does a newer version of R2V allow data to be saved as shapefiles and to be georeferenced? In that case, why did you not use the newer version?
A: We used the version of R2V that was accessible to us at the time of data processing. As for the map sheets which were added to the revised version of the manuscript we used the function of automatic feature generation available in the ArcScan, an extension of ArcGIS. It proved to be a faster method, although required more data editing.

R: Line 154: resultant -> resulting
A: Corrected.

R: Line 161: Remove '(northern hemisphere, zone 33)'
A: Corrected.

R: Line 162: Remove 'later in this work'
A: Corrected.

R: Line 165: GRID -> grid
A: Corrected.

R: Line 168: Remove 'with the working name DEM IGF 1961'
A: Corrected.

R: Line 171: Add reference
A: Corrected.

R: Line 174-186: Make this into one paragraph, and remove the sentence starting 'Work began with the correction of sheet 8…'
A: Corrected.

R: Line 180: Change to 'the location**s** of the elevation points **were** assessed'
A: Corrected.

R: Line 180: Remove 'as many as' and add 'elevation points' after 95
A: Corrected.

R: Line 181: about 50 elevation points? Not exactly 50, which are the same as 50 of the points in the map sheet?
A: Since we added 4 more map sheets, we updated the information and rephrased this sentence in the revised manuscript.

R: Line 187-188: shifted southeastwards by how much?A: As explained before, the shift between two maps was not constant in space. In order to clarify this, we will provide exact value of the mean shift between these sheets.

R: Line 193: (Barna and Warchol, 1987)
A: Corrected.

R: Line 195: Remove 'because much of it was covered by the Greenland Sea'
A: Corrected.

R: Line 202: fragment -> portion
A: Corrected.

R: Line 205: adding a few **points**
A: Corrected.

R: Line 228: Remove 'natural' before 'processes'
A: Corrected.

R: Line 230: Remove 'for obvious reasons'
A: Corrected.

R: Line 231: Change to 'After **considering** the aforementioned criteria, the **part** of the IGF PAN model selected'
A: Corrected

R: Line 247: Change to 'The measure for examining the **extent** and pattern of glacier **retreat**'
A: Corrected

R: Line 248: The research -> This analysis
A: Corrected

R: Line 251: Change to '**During** the study period'
A: Corrected

R: Line 255: most intense -> greatest
A: Corrected

R: Line 256: Change to '**For** these glaciers'
A: Corrected

R: Line 257-258: Rephrase sentence to follow better after the previous one
A: We moved these sentences to the section Discussion.

R: Line 258-261: This comparison to the LIA does not belong in Results, but rather Discussion
A: We moved these sentences to the section Discussion.

R: Line 261-262: Repetition of narrowing of the lowest/lower parts of the glaciers. Rephrase
A: We moved both sentences (lines 257-261) to the discussion section, and therefore this repetition does not occur here in the revised manuscript.

R: Line 268: shrank -> decreased
A: Corrected.

R: Line 268: amounted to -> was
A: Corrected.

R: Line 273: situation -> setting
A: Corrected.

R: Line 281: Change to '**at** their termini'
A: Corrected.

R: Line 286-287 Change to 'the area of Nordfallbreen decreased by only 0.03 km$^2$ (3.6%)**, which is** among the lowest values in the entire region'
A: Corrected.

R: Line 289: Remove one of the 'However'
A: Corrected.

R: Line 290: Space missing after 'zone'
A: Corrected.

R: Line 292: 'receded by only 120 m' - Is not 120 m in 30 years a lot?
A: It is a lot, but this value is still relatively small among the glaciers in the area, and the smallest if we compare it with the retreat rate for the glaciers of similar size. Nevertheless, since it might be confusing we removed the word 'only'.

R: Line 297: shrinkage -> areal decrease
A: Corrected

R: Line 298: Remove 'though this did vary between glaciers'. Change to '**For** the region's largest glaciers'
A: Corrected

R: Line 299: shrinkage -> decrease
A: Corrected

R: Line 299: leading into -> calving into
A: Corrected

R: Line 300: for the glaciers instead of in the glaciers
A: Corrected

R: Line 309: Could add Nuth et al. (2007, 2013) and Holmlund (2021)
A: Corrected

R: Line 313: Specify which modern methods. Could also cite Girod et al. (2018) and Holmlund (2021).
A: We changed it to "structure-from-motion (SfM) photogrammetry" and added both references.

R: Line 315-316 Change to 'The accuracy of simulations prognosing changes in glacier volumes based on dynamics models depends largely on **that** those models hav**e** been initialised correctly'
A: Corrected

R: Line 323-324: How did the temperature change during this period? Relate this to meteorological data to explain the changes.
A: Following previous suggestions in the revised version of the manuscript we are discussing the changes in the temperature and the precipitation in the section "Study area". Answering the question: what concerns the data for the period of 1960-90 – in the nearest station (Polish Polar Station) meteorological measurements were not carried out until the 1970s, and therefore we can only refer to the dataset from Longyearbyen airport collected since the beginning of the 20th century (Nordli et al., 2014). According to Førland et. al. (2011) the mean annual temperature at the Svalbard airport in the period of 1966-1988 increased by 0,52°C/decade, and in the following period (1988-2011) this trend continued with an increase of the mean annual temperature to 1,25°C/decade. For Hornsund a similar trend was observed. In the period of 1971-2000, the mean annual temperature was -4,7°C which in the following years (2001-2015) increased by 1,9 °C.

R: Line 325-326: For the same time period (1961-90), or something else?
A: We could not find an estimate for the same period, Błaszczyk et. al. (2009) refers to the period of 2000-2006.

R: Line 331-332: extending upwards to -> reaching.
A: Corrected

R: Line 354-355: Change to 'Nordf**a**llbreen is adjacent to Nordfallet (824 m a.s.l.) to the south, which shades it against the sun while also providing it additional supply by avalanche**s**'
A: Corrected

R: Line 359: What's the elevation of the equilibrium line?
A: In the western part of the peninsula (except for the ice-free mountains in the northwest) it was about 300-400 m a.s.l. (Jania, 1988). We added this information to the revised manuscript.

R: Line 367-373: Remove 'Ignorance of the principles on which they were compiled may lead to conclusions drawn as to the glacier recession rate being erroneous and, consequently, recession being overestimated for the years 1984–90, as the apparent status in 1984 would be contrary to reality. Specifically, the misapprehension lies in the fact that, a'. Instead, start the second sentence 'Although the IGF PAN field campaign was conducted in the early 1980s, the maps published after the expedition were based on elevation data taken from aerial photos from 1961, upon which only glacier extents were updated (with a change in colour of contours). Crucially, contour lines were not updated in this 1984 edition, and continued to represent the elevations of 1961.'
A: Corrected

R: Line 384-385: Remove last sentence
A: Corrected

**Tables**

R: Table 1: Add a third column with references to the published works from these overflights (or surveys?). Could also add a map next to the table, showing the coverage for each of the flights?
A: Corrected

Tables 2-5: Add references to IGF PAN and NPI data. In Table 2: Vestre Zdanovfjellet with a capital V
A: Corrected

Table 7: It says land-terminating glaciers instead of calving glaciers in the caption
A: Corrected

**Figures**

R: Figure 1: Why the red frame in (c)? Make (c) only cover the area of interest or make the text 'Area of research' in red and put inside red box. Figure could also benefit from having a map showing Svalbard's location in the North Atlantic instead of panel (b)? Add all names mentioned in the text.
Figure 2: Label the maps (a), (b), and (c), and provide information on which sheet is which. Consider showing all ten map sheets, to help illustrate why these three are the most valuable (supported by an explanation in the text).
Figure 3: Mark the extent of this figure in Figure 2 or present them (Figures 2 and 3) together.
Figures 5-8. Merge these figures into one (or at least Figures 5, 6 and 8), and increase the letter size. Add info on contour line interval.
Figure 7: Stupprygen -> Stupryggen. This typo reoccurs in several figures and in the text.
Figures 10-12: Merge into one figure, potentially also adding the plot from Figure 13.
Figure 9 and 14: Present these figures together, to make them easier to compare. Could even present them together with Figure 4.
A: Since the revised version of the manuscript concerns much bigger area all figures needed an update, but we introduced also changes according to your suggestions, which we appreciate.

---

## Author Response (AR1)

Dear Dr Reinhard Drews,

Please find attached a list of changes introduced in the revised version of the manuscript titled *Unlocking archival maps of the Hornsund fjord area for monitoring glaciers of the Sørkapp Land peninsula, Svalbard.*
Kind regards,

Justyna Dudek

**1. Introduction**

In the introduction we have made all changes as suggested by both reviewers.

We removed word glacier after all glacier names. We also merged some paragraphs as suggested by the second reviewer.

We rephrased the first paragraph and added relevant references (Nordli et al., 2014; Isaksen et al., 2016, Nuth et al. 2010; Morris et al., 2020; Schuler et al. 2020).

We merged the second and the third paragraph and added references supporting our statements (Jakob et al, 2012; Nuth et al. 2013; Martín-Moreno et al., 2017).We also rephrased some sentences as per suggestion of the second reviewer.

In the fourth paragraph of the reviewed version (which was merged from the sixth and seventh paragraphs of the preprint) we corrected two sentences as suggested by the second reviewer (line 44 and 47 of the preprint).

In the sixth paragraph of the reviewed version we rephrased first two sentences and added one citation (Kolondra 2005). We also rephrased the last sentence according to the suggestions of the first reviewer.

In the first table we added a third column with references to the published works from the overflights caried out by the NPI (over 40 citations). We also added one figure showing a series of 9 maps of Sorkapp Land lined up side by side showing the regions covered by each photogrammetric mapping campaign.

In the last paragraph we introduced some changes as suggested by the second reviewer (line 68, 69, 73, and 74 of the preprint).

**2. Study area**

In the beginning we added two paragraphs with general information about Svalbard, the influence of the sea currents, strong climate gradients, ice cover of the archipelago. We also introduced the concept of surging glaciers. We added some information about climate (temp. and precipitation change) in southern Spitsbergen. We supported this section with 8 additional citations including one suggested by the reviewer (Hagen et al., 1993; Eckerstorfer and Christiansen, 2011; Farnsworth et al., 2016 ; Sund et al., 2009; Ziaja and Ostafin 2015; Pälli et al., 2003; Grabiec et al., 2017; Isachsen et al 2016, Forland et al., 2011; Osuch I Wawrzyniach 2017).

In the next paragraphs we introduced all corrections suggested by both reviewers (lines 76-79, 85, 88) while also updating the information so that it refers to the larger research area and more glaciers included in the reviewed version of the manuscript.

The figure 2 (previously 1) is now updated covering larger area, and having an additional map showing Svalbard's location in the North Atlantic.

**3. Source material**

Instead of dividing this chapter into two sections describing the data representing a specific period (1961, 1990), we decided to change its order and divide this chapter by data type (maps, images, dems), which we then ordered from the oldest to the newest.

In the first section (3.1. Maps) we introduced most of the suggested changes referring to the data from 1961. The corrected text from section "3.1. 1961 data" (lines 95-139 of the preprint) is now in the section "3.1.1 IGF PAN topographic map series". From the 1961 cartographic data series we added 3 more map sheets representing glaciers and overlapping with the elevation data from 2010. We show their position on the figure 3.

Both figures 2 and 3 in the preprint showing the original IGF PAN maps are now merged and their content is shown on the figure 3 of the reviewed version of the manuscript.

We added the section "3.1.2 NPI map" detailing the specification of the online map released by NPI and representing the year 1990 which we used in our research (lines 175 to 182). I this section we also added Figure 4 showing the extent of the data for 1990 and 1961 on this map.

We added the section "3.2. Imagery" I which we described data from 3 photogrammetric campaigns (1961, 1990, and 2010) and Landsat 5 TM scenes used in our research. Figure 5, added to reviewed version is showing their spatial coverage.

The last section of this chapter "3.3. Dems" added to the corrected manuscript contains information about elevation datasets released by NPI. Added figure 6. shows their extent. Important change in this section: four research we added data from 2010 which constituted our baseline dataset throughout the study.

**4. Methods**

The most important change in this chapter refers to the data extent and the tables with coordinates of the topographic points. As mentioned above we decided to add more map sheets, and in order to avoid having too many tables in the text we decided to merge the information from the tables 2-4 together with the same information for additional map sheets and add it to the manuscript as a supplement. In this chapter we decided to keep mostly the information regarding the processing of the maps while the description of the changes in glacier extent was moved to added chapter " 5. Results". We also changed the division into sections in this chapter and in the reviewed manuscript some of the sections has different names and content. In the corrected version of the manuscript we have 4 sections:

**4.1 Source data processing and evaluation of output data accuracy** (previously in the preprint 4. Methods of source-data processing and evaluation of output data accuracy and 4.1 Verification of source data accuracy);

The biggest change described in this section is the work flow for processing the scanned maps. In the corrected version we decided to do the data processing using only the tools available in Arc Scan extension of ArcGIS software. For the standardization of data processing methods we also repeated the processing for the data presented in the preprint. In this section we also added Figure 7 showing our initial work flow.

Figure 8. (previously 4) shows larger study area.

In this section there is also the description of the three additional maps sheets. All maps sheets are presented on the figures 9-14 (previously 5,6, and 8). Although previously we planned to plot vector arrows pointing in the direction of the dx, dy offset to the NPI reference map we didn't manage to do it efficiently because they were not clearly visible in the adopted scale.

**4.2 Fitting data from 1961 and 2010** (previously in the preprint it was the same title)

We updated the section with regards to comparison of 1961 data against 2010 NPI model.

**4.3 Final elevation model for 1961**

We introduced some corrections suggested by reviewers (lines 228, 230, 231, 242 of the preprint). We updated all values relevant bigger study area. Both figures 16 (previously 9) and 20 (previously 14) were updated with new dataset for larger area compared against the data for 2010.

**4.4 1961–1990–2010 changes in glacier geometries**

We left in this section only 3 phrases referring to methods for determining glaciers geometry changes.

5. **Results**

In this chapter we included 2 tables with data about surface area and surface area change of each analysed glacier in the periods 1961-1990, 1990-2010, and 1961-2010. We also introduced all changes suggested by the reviewers (lines of the preprint: 251, 255, 256, 257-258, 258-262, 268, 273, 281, 286-287, 289, 290, 292, 297, 298, 299, 300). We removed fig 16 of the preprint. We described the results adding information about glacier changes in the period 1990- 2010 (or updating the values for the period 1961-2010). The figure 21 (previously 15) was updated showing larger area extent and additional research periods.

6. **Discussion**

We updated the research periods in the text, added references as suggested by the reviewers, and make all corrections as per line by line comments.

---

## Author Response (AR2)

Dear Dr Reinhard Drews

Please find below our answers to the reviewers comments, which summarizes the mpovements made to our manuscript.

Kind regards,

Justyna Dudek

Answers to the first review:

R: Dudek & Petlicki digitize and co-register a set of six topographic map sheets from the Institute of Geophysics of the Polish Academy of Sciences (IGF PAN) covering Sørkapp Land (the southern tip of Svalbard) from 1961. Dudek & Petlicki then compare the digitized 1961 elevation data to DEMs created by the Norwegian Polar Institute for 1990 and 2010. They conclude with some observations and hypotheses for what factors drove the observed changes in glacier thickness and extent between 1961 and 2010. Overall, I think this manuscript is much improved from the original submission. The text is easy to understand, and the figures are illustrative and well-done. I hope that my general, specific, and technical comments below will help the authors improve the manuscript.

A: We appreciate your valuable feedback and we corrected our manuscript (the text and figures) according to these suggestions. We believe that they helped substantially to improve the quality of our work. Below we included our answers. Referee text (R) and author responses (A) are indicated.

General comments

R: Figure 1 (illustrating the coverage of photogrammetric campaigns from the Norwegian Polar Institute) indicates that there is 100% complete coverage of Sørkapp Land from air-photos in 1960 and 1961. Given the availability of this imagery and the fact that there are now tools (e.g., Agisoft, MicMac) to make structure-from-motion photogrammetry a relatively fast and effective strategy for 3D reconstructions of Svalbard glaciers (Mertes et al., 2017; Midgley and Tonkin, 2017; Girod et al., 2018; Holmlund, 2021; Geyman et al., 2022), are you able to justify why digitizing contour lines from maps (constructed from more qualitative photogrammetric approaches) is the approach you take, rather than making the 3D model directly from the 1961 imagery?

A: We ran some trials on the images we possess (using Agisoft software), unfortunately without satisfying outcomes, due to the low-quality version of our photos. Delivery of a high-quality output dataset requires a high-quality input dataset, which we do not have access to.

R: Based on the description in section 4.2, it sounds like you made a DEM from the digitized 1961 contour lines (TIN interpolation of the vector contours), and then differenced that DEM to the modern (2010) NPI DEM. It would be much better to instead difference the 1961 contour lines to the 2010 DEM, and then interpolate these lines of 1961-2010 *difference* values. Finally, you would add this interpolated difference map back to the 2010 DEM in order to recover the 1961 DEM. The reason for doing the DEM difference this way is that ice

elevation changes are much smoother in space than topography is. For example, if the ice melted -50 meters during the period of interest in one place and -60 meters at a nearby location, it is likely that the ice elevation change halfway between is about -55 meters. In contrast, if the elevation at one point is 600 meters, and the elevation at a nearby point is 700 meters, it is far less likely that your guess of 650 meters for a point halfway in between is accurate.

A: Thank you for this advice. In our revised version of the manuscript we differentiated 2010 DEM directly from contour lines (converted to points with 5 m spacing along the lines), and we obtained our 1961 DEM by subtracting difference values from 2010.

Specific comments

R: Lines 16-17: You may want to add a reference after this sentence. Nordli et al. (2014) or Nordli et al. (2020) would work well.

A: We decided to add  Nordli et al. (2020) to the lines 17-18, and  Nordli et al. (2014)  was already cited in that paragraph.

R: Lines 63-66: You might consider citing Geyman et al. (2022) here, since they provide 3D photogrammetric reconstructions of all the glaciers in the area from the Norwegian's 1936 air photo archive.

A: Added

R: Lines 70-71: "No other set of data of the same spatial extent was created until the year 2010 (Fig. 1)." -> See above comment—this isn't true, as there are published 1936 reconstructions of glacier extents and volumes.

A:   Although the data for 1936 is very important for reconstructions of glacier extents after the end of LIA it consists of oblique images that have some limitations (i.e. in presenting parts of the glaciers hidden behind the high peaks). In this sentence, we rather meant that after 1961 we had to wait almost half of the century for another data set that would be comparable in coverage and uniformity to the data from 60. (namely vertical images covering uniformly all glaciers in Sørkapp Land).

R: Table 1: Note that Geyman et al. (2022) covers the entire peninsula in 1936.

A: Added

R: Figure 16: Since the colorbar on the image spans such a large range (-100 to +100 m), it would help the reader if you point to the broad area of light pink in Breinesflya and wrote what the average delta z is for that region. Is it -5 m, -25 m, etc.?

A: We changed the range of the colorbar on the images for areas used to validate DEM to (-50 to +50 m). For glaciers we kept a range (-100 to +100 m). For Breinesflya average delta z does not exceeds -5 m.

R: Lines 301-302: "may result from processes going on in the natural environment, e.g. melting of dead ice in marginal zones of glaciers…" You might want to rephrase this sentence, because the melting of dead ice on the marginal zones of glaciers is still of interest to all of the scientists interested in quantifying the ice loss and contribution to sea level rise from Svalbard.

A: Yes, you are right, these zones are very dynamic and this is the reason why they shouldn't be considered as 'stable areas' suitable as a reference surface for DEM validation. Our final version of DEM can be used for quantifying the melting of dead ice.

R: Lines 313-314: "The mean elevation difference (the bias) between the compared models was 2.28 m, with a standard deviation of 3.18 m, indicating that the 1961 model is higher." Another important statistic for you to report is the mean absolute error – take the absolute value of the delta z map, and then compute the mean. This gives the reader a good sense of the spread of the data.

A: In our revised manuscript we performed corrections for each sheet separately, and we obtained different results.

R: Figure 20: Something that makes me nervous about this delta z map is that there seems to be a systematic trend with elevation: the low elevation regions tend to have negative delta z, and the high elevation regions tend to have positive delta z. I recommend adding a figure to show this pattern. Do you have ideas for why it appears, or ways to fix it?

A: These differences does not exceed 5 m (a values falls in the accuracy defined for the reference dataset for 2010). In the revised version of the manuscript we decided to show differences in unglaciated area between the years 1990 and 2010, both extracted by NPI directly from aerial photographs and we observed that each dataset we used in manuscript have regions of positive or negative delta z, regardless of methods used for data production.

Line 319: "The measure for examining the extent and pattern of glacier retreat in the years 1961–1990–2010 was changes in their surface area, the rate of frontal recession and – where data allowed (i.e. for land-based glaciers) – changes in thickness." What do you mean by "land-based glaciers"? All glaciers on Svalbard are land-based. The ice also is all grounded. Additionally, there are bathymetry compilations of Svalbard's fjords, so you should be able to compute glacier mass loss for all glaciers in your region, regardless of whether they terminate on land or in the ocean. Since you address this question partially in lines 456-460, you might consider moving that paragraph here (e.g., line 319).

A: In this context by the "land-based" glaciers, we meant the glaciers terminating on land and for better understanding, we changed that term to "land-terminating" in the entire manuscript. For marine terminating glaciers (referred to as "tidewater" glaciers in our text) we are lacking some data that has been removed from the IGF PAS maps - namely the thickness of glacier fronts. In the newest version of our manuscript we added Table 5 with information about thickness change for each glacier and for the whole glaciated area.

R: Lines 329-337: I think this paragraph would be well supported by a figure that plots 1961-2010 elevation change (delta z) on the y-axis vs. altitude on the x-axis (you could use either the 1961 or the 2010 elevation to define this axis, just specify). That way, readers can see

how elevation (and therefore, through the adiabatic lapse rate, temperature) affects mass balance.

A: This is very good suggestion, but for final version we decided to reduce the number of figures and table to improve readability of the manuscript.

R: Figure 21: the blue (positive delta z) area in the upper reaches of Mendeleevbreen looks suspicious to me. Have you done any tests to make sure that there isn't a mistake with the georeferencing or warping there?

A: Mendeleevreen it is a surging glacier. In the period of 1961-1990 it was in a quiescent phase, with building up of accumulation zone. Glacier surged in 2004, and if we compare the elevation dataset from the year 2004 (for example ASTER DEM) with the data for 2010 we can clearly see that a "build up" was at the front of the glacier, which indicated surging phase. Moreover NPI dataset for 1961 which also covers that glacier, shows similar patern.

Lines 461-463: are you able to report what the average dh/dt was during 1961-1990 for all of the valid glacier area in your region? That is a number that would interest many glaciologists.

A: We now included this information for the period 1961-2010 together with dh for each glacier in the table 5.

Line edits (typos and technical corrections)

Line 2: "their geometry" -> "glacier mass, extent, and geometry"

A: Corrected

Line 2: "them" -> "archival maps"

A: Corrected

Line 5: "The research objective" -> "the objective of this research"

A: Corrected

Line 7: "in in" -> "in"

A: Corrected

Line 96: add parentheses to the Isaksen et al. (2016) reference.

A: Corrected

Line 115: "To the east, it…" -> "To the east, Samarinbreen…"

A: Corrected

Lines 139-140: "consisted primarily of data presenting topographic surface (topographic maps and DEMs), supplemented with imagery (aerial photos and satellite images)" ->

"consisted primarily of topographic maps and DEMs, supplemented with aerial photos and satellite images"

A: Corrected

Line 156: Delete "to elaborate results (especially on changes in glacier thickness)"

A: Corrected

Line 157: Do you mean "level of accuracy" rather than "specificity"?

A: We rather meant "specifity", but it does relate to the level of accuracy as well.

Line 160: Delete "prepared"

A: Corrected

Line 164: Use a consistent dash width between 1961-1990-2010.

A: Corrected

Lines 189-190: Delete "Their specification is provided below (Fig. 5)" and put the "(Fig. 5)" reference at the end of the previous sentence.

A: Corrected

Line 428: Delete "in places"

A: Corrected

Line 454: Use a consistent dash width between 1961-1990-2010.

A: Corrected

References
Geyman, E.C., JJ van Pelt, W., Maloof, A.C., Aas, H.F. and Kohler, J., 2022. Historical glacier change on Svalbard predicts doubling of mass loss by 2100. Nature, 601(7893), pp.374-379.

Girod, L., Nielsen, N. I., Couderette, F., Nuth, C., and Kääb, A.: Precise DEM extraction from Svalbard using 1936 high oblique imagery, Geosci. Instrum. Method. Data Syst., 7, 277–288, https://doi.org/10.5194/gi-7-277-2018, 2018.

Holmlund, E. (2021). Aldegondabreen glacier change since 1910 from structure-from-motion photogrammetry of archived terrestrial and aerial photographs: Utility of a historic archive to obtain century-scale Svalbard glacier mass losses. Journal of Glaciology, 67(261), 107-116.

Mertes, J. R., Gulley, J. D., Benn, D. I., Thompson, S. S., Nicholson, L. I.: Using structure-from-motion to create glacier DEMs and orthoimagery from historical terrestrial and oblique

aerial imagery, Earth Surface Processes and Landforms, 42(14), 2350-2364, 2017.

Midgley, N. G., Tonkin, T. N.: Reconstruction of former glacier surface topography from archive oblique aerial images, Geomorphology, 282, 18-26, 2017.

Nordli, Ø., Przybylak, R., Ogilvie, A., Isaksen, K.: Long-term temperature trends and variability on Spitsbergen: The extended Svalbard Airport temperature series, 1898–2012, Pol. Res., 33, 21349 (2014).

Nordli, Ø. et al. Revisiting the extended Svalbard Airport monthly temperature series, and the compiled corresponding daily series 1898–2018. Polar Res. (2020).

A: We added all suggested references to our corrected manuscript

Answers to the second review:

This is my first review of this study by Dudek and Petlicki. I carefully reviewed the revised manuscript as well as the previous reviews and replies by the authors. I unfortunately could not find a way to access the original manuscript to assess the progress since the review. Overall, this study is relatively well written, the methods are sound and the results seem satisfactory. The study is interesting as most studies working with archival topographic maps, that I am aware of, usually address the issues related to this data in a very synthetic way (e.g. Ye et al., 2015; Nuimura et al., 2012). However, I share the impression of the previous reviewers that the method is not extremely novel (see main comment below).

A: We appreciate your valuable feedback and we corrected our manuscript (the text and figures) according to these suggestions. We believe that they helped substantially to improve the quality of our work. Below we included our answers. Referee text (R) and author responses (A) are indicated.

General comments:
- Methods and methods description:

R: I stand with R1 and R2 who highlight that the methods need to be clearly presented and novel enough for this study to be a benchmark for other studies analyzing topographic maps. I also do not find the methods extremely novel, but the results seem satisfactory anyway. There are two ways of improvements though, that have already been brought up by the previous reviewers that could be considered:
Interpolate the elevation difference map (DoD) rather than interpolate the contour lines with TIN. This was brought up by R1 and considered to be addressed by the authors, but I noticed that the latter method is still used in this version. I quickly checked the data available on the Zenodo repository, and the slope derived from the provided 1961 DEM shows interpolation artifacts. This could probably be improved with R1's suggestion.

A: Thank you for this suggestion. In order to minimize the impact of DEM artifacts in our latest revised version we used contour lines only (converted to points with spacing 5 m) and we interpolated the elevation difference data (DoD) instead of contour lines with TIN/ grid. We obtained much better results, nevertheless it has to be noted that new method is much more time consuming and requires more memory and disk space for data processing.

R: There are still relatively large and systematic elevation differences in stable terrain (e.g., in the west of the study area on Figure 20). It looks like the elevation difference shows a step-like pattern at the junction of two sheets. I see at lines 313-314 that you correct for a mean bias, but if I understand correctly, you calculate a single bias for the DEM mosaic of all sheets at once? If so, I would recommend estimating this bias for each sheet individually. If this does not work, I would encourage you to follow the suggestion of R2 to apply a blockwise coregistration. I understand your preference to use the topographic point for the horizontal alignment, but you could still apply this method to correct for a smooth vertical offset.

A: In the first version of our manuscript we calculated bias for a mosaic of all sheet at once. In revised version we performed correction for each map sheet separately. As a first step we calculated mean shift for each map using algorithm developed by Nuth and Kääb in 2011. Second step consisted of feature adjustment (for each sheet separately), correction for Z and then merging all the data. Between two map sheets 8 and 10 there was some small step at the junction and we diminished this error, although it is still visible. Since elevation bias for Breinesflya at the map sheet 10 does not exceeds ~ 5 m we decided to accept that. We also observed some small steps when comparing the data from NPI for 1990 and 2010 which we use as a reference dataset and we believe that currently the elevation data for Sorkapp Land that would not contain some small errors does not exist.

R: I would also include in the text some parts of your replies to the reviewers, especially concerning 1) your unsuccessful attempt to process low quality scans of the images unsuccessfully 2) your attempt of applying the Nuth & Kaab (2010) method to all stable terrain and the issue of non-rigid transformation (i.e., a shift alone is not enough).

A: In revised version of our manuscript we now mentioned Nuth & Kaab (2011). We processed each map using this method, and we estimated differences between the data from 1990 and 2010 published by NPI.

R: Finally, I believe that the methods would need to be more detailed to make this work fully reproducible.
Figure 7 provides a nice list of the different steps and commands used in ArcGIS. I have never worked with ArcGIS, however, I assume that there are several tools e.g., in the "ARC SCAN/VECTORISATION" box and several options for each tools. It would be extremely useful to provide the exact tools and options that were used (e.g., in a supplementary table if this takes too much space) and whenever possible, provide a clear description of the algorithms, so that this could be reproduced with open-source alternatives. For example, what are the steps in the "RASTER CLEANUP" or "FEATURE CLEANUP"?

A: In the first version of our manuscript we added more detailed description of tools we used for data processing and we removed them as per suggestion of the reviewers. For the second revision we added more information to the figure listing processing steps, but it is still slightly less than in the preprint.

- Results

R: It is a shame that the authors spend so much effort in generating a DEM from the archival map, but the only results that are presented are only vague numbers of maximum thinning

and a map. I don't think it would be much more effort to calculate an actual glacier-wide mass balance or at least a mean elevation change for each glacier and for each period. This could then be compared to existing estimates in the literature for this area of the whole of Svalbard and make this study more valuable.

A: Good suggestion. In the revised version of our manuscript we added information about mean elevation difference for glaciers.

R: I also wondered why you did not use the information on the 1984 map to generate glacier contours for 1984. You said that the elevation contour lines were not updated, but the color were changed to reflect glacier area changes. With this information, you could include a fourth period in your tables.

A: We decided to focus our research on the years for which we have both: information about area and elevation of glaciers.

- Overall structure
R: Although the manuscript reads relatively well, there are still some parts that could be better structured to be more easily followed, or some sections that are not completely appropriate.
The introduction lacks acknowledgment of recent studies using archival terrestrial and aerial images: Girod et al. (2018), Holmlund et al (2021), Geyman et al. (2022). The first two are briefly mentioned in the discussions, but definitely belongs to the introduction as they are part of the state-of-the-art. The last one is very recent and not yet referenced but would deserve to be acknowledged as well.

A: In revised version we added all three suggested references

R: The Results and Discussion sections are not so well structured. The use of subsections could help guiding the reader. There sometimes seem to be repetitions. Some of it stems from the fact that you first describe land-terminating glaciers then marine terminating glaciers, but this is not always very clear.

A: Thank you for this suggestion.

R: The conclusions should summarize the main findings and methods. Instead, the current conclusions enter too quickly in very specific details of the study, such as the map date. It also does not summarize the findings on area changes. Hence, I suggest rewriting the conclusions.

A: Our main aim was the description of data derived from old maps, therefore we focused on data processing, quality, and its applicability in polar research. We aimed to reflect that goal also in our conclusions.

Specific comments:

R: l 30: you could add references to the studies by Girod et al. (2018) or Geyman et al. (2022).

A: Added

R: L 52-54: Mannerfelt et al. (2022), already referenced elsewhere, is another good example of use of historical terrestrial images in Svalbard.

A: We added this reference in line 34 of the manuscript, where is more relevant.

R: L 85: Maybe the currents that you discuss could be represented on Figure 2?

A: After trying to fit them on the figure we decided to leave it as it is.

R: Table 1, line 1936. You could include the references to Girod et al. (2018) and Geyman et al. (2022) in the list of references.

A: We added Geyman et al. (2022) and the work of Girod et al. (2018) was omitted in the table since it did not concern the peninsula itself (nevertheless we cited it elsewhere in the manuscript).

R: L 96: the reference does not have the right format

A: Corrected

R: Figure 2: This figure should probably appear first and be referenced at the first mention of the Sorkapp Land peninsula. Could you please indicate the source of the background map?

A: Corrected

R: L 139: "analyzes", do you mean "analysis"?

A: Yes, corrected.

R: L 183: remove "therefore" as it is redundant with "since".

A: Corrected

R: Figure 7: The figure could be improved to include the inputs and outputs. Also see my comment on detailing the individual steps a bit more.

A: We developed the figure by adding several steps.

R: L 239 "The six maps shared 189 points representing the same places" I understand from this sentence that the different sheets have some overlap. What is the size of this overlap (in map units)? Could this not be used to first align the sheets relatively to each other? This would be particularly useful for sheets with little topographic points or stable terrain.

A: The map sheets from IGF PAS representing the year 1961 do not overlap. In this sentence, we meant that we found the same points in the datasets from 1990/2010, and based on these points we performed co-registration of the data from 1961.

R: - L 286: "The vector data thus processed was then used to generate a DEM" You may want to briefly re-state how you derive the DEM, or refer to the appropriate subsection. It took me some time to realize it was explained further up.

A: We rephrased this sentence.

R: - Figure 16 and 20: I would adapt the color scale for these figures to better show the residuals. A min/max value of about -50/50 or less would probably be more appropriate.

A: Corrected

R: L 313-314: "The mean elevation difference (the bias) between the compared models was 2.28 m, with a standard deviation of 3.18 m, indicating that the 1961 model is higher.". Could this bias correction be done individually for each sheet? Also it is recommended to use the median rather than mean, because it is less sensitive to outliers (Höhle & Höhle, 2009). The 3.18 m standard deviation seem surprisingly low when looking at figure 20. Maybe the change of colour scale would show it is not. Or maybe it would be useful to display the areas masked?

A: For the final version of the manuscript we added absolute error a suggested by the second Referee

R: Figure 21: The legends are too small to be read.

A: Corrected

R: Table 3: Brackets are lacking in the table (for percentage).

A: Corrected

R: L 381 "insolation thickness" -> "insolation, thickness"

A: Corrected

R: L 389-395: this whole paragraph simply repeats what is in the introduction. This should all be moved to the introduction (as the references).
A: We moved this chapter to the Introduction section

R: L 393: There are a lot more recent and appropriate references for the declassified spy satellite studies, such as Maurer et al. (2015); Maurer et al. (2019); Dehecq et al. (2021), Bhattacharya et al. (2021). You do not need to cite them all of course.

A: Added

R: L 395: you could reference Geyman et al. (2022) here.

A: Added

R: L 400: I believe the reference to Zekollari et al. (2020) would be more appropriate here.

A: Corrected

R: Supplementary: In the supplementary table, you should describe the table content (it should be understandable without searching in the main text) and the meaning of each column, as it is not very clear.

A: Corrected

References:
Bhattacharya, A., Bolch, T., Mukherjee, K., King, O., Menounos, B., Kapitsa, V., Neckel, N., Yang, W., Yao, T., 2021. High Mountain Asian glacier response to climate revealed by multi-temporal satellite observations since the 1960s. Nat Commun 12, 4133. https://doi.org/10.1038/s41467-021-24180-y

Dehecq, A., Gardner, A.S., Alexandrov, O., McMichael, S., Hugonnet, R., Shean, D., Marty, M., 2020. Automated Processing of Declassified KH-9 Hexagon Satellite Images for Global Elevation Change Analysis Since the 1970s. Front. Earth Sci. 8. https://doi.org/10.3389/feart.2020.566802

Geyman, E.C., J. J. van Pelt, W., Maloof, A.C., Aas, H.F., Kohler, J., 2022. Historical glacier change on Svalbard predicts doubling of mass loss by 2100. Nature 601, 374–379. https://doi.org/10.1038/s41586-021-04314-4

Höhle, J., Höhle, M., 2009. Accuracy assessment of digital elevation models by means of robust statistical methods. ISPRS Journal of Photogrammetry and Remote Sensing 64, 398–406. https://doi.org/10.1016/j.isprsjprs.2009.02.003

Maurer, J., Rupper, S., 2015. Tapping into the Hexagon spy imagery database: A new automated pipeline for geomorphic change detection. ISPRS Journal of Photogrammetry and Remote Sensing 108, 113–127. https://doi.org/10.1016/j.isprsjprs.2015.06.008

Maurer, J.M., Schaefer, J.M., Rupper, S., Corley, A., 2019. Acceleration of ice loss across the Himalayas over the past 40 years. Science Advances 5, eaav7266. https://doi.org/10.1126/sciadv.aav7266

Nuimura, T., Fujita, K., Yamaguchi, S., Sharma, R.R., 2012. Elevation changes of glaciers revealed by multitemporal digital elevation models calibrated by GPS survey in the Khumbu region, Nepal Himalaya, 1992-2008. Journal of Glaciology 58, 648–656. https://doi.org/10.3189/2012JoG11J061

Ye, Q., Bolch, T., Naruse, R., Wang, Y., Zong, J., Wang, Z., Zhao, R., Yang, D., Kang, S., 2015. Glacier mass changes in Rongbuk catchment on Mt. Qomolangma from 1974 to 2006 based on topographic maps and ALOS PRISM data. Journal of Hydrology 530, 273–280. https://doi.org/10.1016/j.jhydrol.2015.09.014

Zekollari, H., Huss, M., Farinotti, D., 2020. On the Imbalance and Response Time of Glaciers in the European Alps. Geophysical Research Letters 47, e2019GL085578. https://doi.org/10.1029/2019GL085578

A: Thank you for these suggestions, we cited some of the proposed positions where appropriate.

---

## Author Response (AR3)

Dear Reinhard Drews,

We appreciate your positive feedback on our manuscript and technical comments on how to improve it. We are pleased that the corrections to our revised manuscript have satisfied you and the reviewers. We would like to apologize for the long wait for the final version of our dataset and the manuscript, which was dictated by the extra work we had to put into the added map sheets and different methods for processing them.

What is concerning your remaining comments to the accepted version of the manuscript we added an information about low quality version of our photos which we tried to process. It is placed at the end of the Introduction section. We also corrected misspellings, and changed (or removed) some of the adjectives such as "accurate", "reliable" and "invaluable" (including one word in Abstract).

In addition, we decided to change layouts of the figures 1, 8, and 14 so that they are better placed in the text. Below we list all changes and corrections that we made to the figures:

Figure 1: We changed configuration of the panels from vertical to horizontal layout.

Figure 7: We made some changes in the fonts used for the peak names by removing offsets of their shadow effect.

Figure 8: We changed the configuration of the text boxes so that the figure fits into one column of the manuscript.

Figure 10: We enlarged labels (a)-(d) and changed their placement to the top left corners of each panel of the figure in order to increase their visibility. Similarly like in the case of the Fig. 7 we removed offsets of the shadow effect in the fonts used for peak names.

Figure 14: We changed layout from horizontal to vertical, we added (a)-(c) labels to each panel and changed the caption accordingly.

Once again thank you for your kind response.

Best regards,

Justyna Dudek and Michał Pętlicki